# Scale Up Nonlinear Component Analysis with Doubly Stochastic Gradients

**Bo Xie[1], Yingyu Liang[2], Le Song[1]**
[1]Georgia Institute of Technology
bo.xie@gatech.edu, lsong@cc.gatech.edu
[2]Princeton University
yingyul@cs.princeton.edu

## Abstract

Nonlinear component analysis such as kernel Principle Component Analysis (KPCA) and kernel Canonical Correlation Analysis (KCCA) are widely used in machine learning, statistics and data analysis, but they cannot scale up to big datasets. Recent attempts have employed random feature approximations to convert the problem to the primal form for linear computational complexity. However, to obtain high quality solutions, the number of random features should be the same order of magnitude as the number of data points, making such approach not directly applicable to the regime with millions of data points.

We propose a simple, computationally efficient, and memory friendly algorithm based on the "doubly stochastic gradients" to scale up a range of kernel nonlinear component analysis, such as kernel PCA, CCA and SVD. Despite the *non-convex* nature of these problems, our method enjoys theoretical guarantees that it converges at the rate $\tilde{O}(1/t)$ to the global optimum, even for the top $k$ eigen subspace. Unlike many alternatives, our algorithm does not require explicit orthogonalization, which is infeasible on big datasets. We demonstrate the effectiveness and scalability of our algorithm on large scale synthetic and real world datasets.

## 1 Introduction

Scaling up nonlinear component analysis has been challenging due to prohibitive computation and memory requirements. Recently, methods such as Randomized Component Analysis (RCA) [12] are able to scale to larger datasets by leveraging random feature approximation. Such methods approximate the kernel function by using explicit random feature mappings, then perform subsequent steps in the primal form, resulting in linear computational complexity. Nonetheless, theoretical analysis [18, 12] shows that in order to get high quality results, the number of random features should grow linearly with the number of data points. Experimentally, one often sees that the statistical performance of the algorithm improves as one increases the number of random features.

Another approach to scale up the kernel component analysis is to use stochastic gradient descent and online updates [15, 16]. These stochastic methods have also been extended to the kernel case [9, 5, 8]. They require much less computation than their batch counterpart, converge in $O(1/t)$ rate, and are naturally applicable to streaming data setting. Despite that, they share a severe drawback: all data points used in the updates need to be saved, rendering them impractical for large datasets.

In this paper, we propose to use the "doubly stochastic gradients" for nonlinear component analysis. This technique is a general framework for scaling up kernel methods [6] for *convex problems* and has been successfully applied to many popular kernel machines such as kernel SVM, kernel ridge regressions, and Gaussian process. It uses two types of stochastic approximation simultaneously: random data points instead of the whole dataset (as in stochastic update rules), and random features instead of the true kernel functions (as in RCA). These two approximations lead to the following

benefits: (1) **Computation efficiency** The key computation is the generation of a mini-batch of random features and the evaluation of them on a mini-batch of data points, which is very efficient. (2) **Memory efficiency** Instead of storing training data points, we just keep a small program for regenerating the random features, and sample previously used random features according to pre-specified random seeds. This leads to huge savings: the memory requirement up to step $t$ is $O(t)$, independent of the dimension of the data. (3) **Adaptibility** Unlike other approaches that can only work with a fixed number of random features beforehand, doubly stochastic approach is able to increase the model complexity by using more features when new data points arrive, and thus enjoys the advantage of nonparametric methods.

Although on first look our method appears similar to the approach in [6], the two methods are fundamentally different. In [6], they address *convex problems*, whereas our problem is highly *non-convex*. The convergence result in [6] crucially relies on the properties of convex functions, which do not translate to our problem. Instead, our analysis centers around the stochastic update of power iterations, which uses a different set of proof techniques.

In this paper, we make the following contributions.

**General framework** We show that the general framework of doubly stochastic updates can be applied in various kernel component analysis tasks, including KPCA, KSVD, KCCA, *etc.*.

**Strong theoretical guarantee** We prove that the finite time convergence rate of doubly stochastic approach is $\tilde{O}(1/t)$. This is a significant result since 1) the *global* convergence result is w.r.t. a *non-convex problem*; 2) the guarantee is for update rules without explicit orthogonalization. Previous works require explicit orthogonalization, which is impractical for kernel methods on large datasets.

**Strong empirical performance** Our algorithm can scale to datasets with millions of data points. Moreover, the algorithm can often find much better solutions thanks to the ability to use many more random features. We demonstrate such benefits on both synthetic and real world datasets.

Since kernel PCA is a typical task, we focus on it in the paper and provide a description of other tasks in Section 4.3. Although we only state the guarantee for kernel PCA, the analysis naturally carries over to the other tasks.

## 2   Related work

Many efforts have been devoted to scale up kernel methods. The random feature approach [17, 18] approximates the kernel function with explicit random feature mappings and solves the problem in primal form, thus circumventing the quadratic computational complexity. It has been applied to various kernel methods [11, 6, 12], among which most related to our work is RCA [12]. One drawback of RCA is that their theoretical guarantees are only for kernel matrix approximation: it does not say anything about how close the solution obtained from RCA is to the true solution. In contrast, we provide a finite time convergence rate of how our solution approaches the true solution. In addition, even though a moderate size of random features can work well for tens of thousands of data points, datasets with tens of millions of data points require many more random features. Our online approach allows the number of random features, hence the flexibility of the function class, to grow with the number of data points. This makes our method suitable for data streaming setting, which is not possible for previous approaches.

Online algorithms for PCA have a long history. Oja proposed two stochastic update rules for approximating the first eigenvector and provided convergence proof in [15, 16], respectively. These rules have been extended to the generalized Hebbian update rules [19, 20, 3] that compute the top $k$ eigenvectors (the subspace case). Similar ones have also been derived from the perspective of optimization and stochastic gradient descent [20, 2]. They are further generalized to the kernel case [9, 5, 8]. However, online kernel PCA needs to store all the training data, which is impractical for large datasets. Our doubly stochastic method avoids this problem by using random features and keeping only a small program for regenerating previously used random features according to pre-specified seeds. As a result, it can scale up to tens of millions of data points.

For finite time convergence rate, [3] proved the $O(1/t)$ rate for the top eigenvector in linear PCA using Oja's rule. For the same task, [21] proposed a noise reduced PCA with linear convergence rate, where the rate is in terms of epochs, *i.e.*, number of passes over the whole dataset. The noisy power method presented in [7] provided linear convergence for a subspace, although it only converges linearly to a constant error level. In addition, the updates require explicit orthogonalization, which

| **Algorithm 1:** $\{\alpha_i\}_1^t = $ **DSGD-KPCA**$(\mathbb{P}(x), k)$ | **Algorithm 2:** $h = $ **Evaluate**$(x, \{\alpha_i\}_{i=1}^t)$ |
|---|---|
| **Require:** $\mathbb{P}(\omega)$, $\phi_\omega(x)$. | **Require:** $\mathbb{P}(\omega)$, $\phi_\omega(x)$. |
| 1: **for** $i = 1, \ldots, t$ **do** | 1: Set $h = 0 \in \mathbb{R}^k$. |
| 2:    Sample $x_i \sim \mathbb{P}(x)$. | 2: **for** $i = 1, \ldots, t$ **do** |
| 3:    Sample $\omega_i \sim \mathbb{P}(\omega)$ with seed $i$. | 3:    Sample $\omega_i \sim \mathbb{P}(\omega)$ with seed $i$. |
| 4:    $h_i = $ **Evaluate**$(x_i, \{\alpha_j\}_{j=1}^{i-1}) \in \mathbb{R}^k$. | 4:    $h = h + \phi_{\omega_i}(x)\alpha_i$. |
| 5:    $\alpha_i = \eta_i \phi_{\omega_i}(x_i) h_i$. | 5: **end for** |
| 6:    $\alpha_j = \alpha_j - \eta_i \alpha_j^\top h_i h_i$, for $j = 1, \ldots, i-1$. | |
| 7: **end for** | |

is impractical for kernel methods. In comparison, our method converges in $O(1/t)$ for a subspace, without the need for orthogonalization.

## 3 Preliminaries

**Kernels and Covariance Operators** A kernel $k(x, y) : \mathcal{X} \times \mathcal{X} \mapsto \mathbb{R}$ is a function that is positive-definite (PD), i.e., for all $n > 1$, $c_1, \ldots, c_n \in \mathbb{R}$, and $x_1, \ldots, x_n \in \mathcal{X}$, we have $\sum_{i,j=1}^n c_i c_j k(x_i, x_j) \geq 0$. A reproducing kernel Hilbert space (RKHS) $\mathcal{F}$ on $\mathcal{X}$ is a Hilbert space of functions from $\mathcal{X}$ to $\mathbb{R}$. $\mathcal{F}$ is an RKHS if and only if there exists a $k(x, x')$ such that $\forall x \in \mathcal{X}, k(x, \cdot) \in \mathcal{F}$, and $\forall f \in \mathcal{F}, \langle f(\cdot), k(x, \cdot) \rangle_\mathcal{F} = f(x)$. Given $\mathbb{P}(x)$, $k(x, x')$ with RKHS $\mathcal{F}$, the covariance operator $A : \mathcal{F} \mapsto \mathcal{F}$ is a linear self-adjoint operator defined as $Af(\cdot) := \mathbb{E}_x[f(x) k(x, \cdot)]$, $\forall f \in \mathcal{F}$, and furthermore $\langle g, Af \rangle_\mathcal{F} = \mathbb{E}_x[f(x) g(x)]$, $\forall g \in \mathcal{F}$.

Let $F = (f_1(\cdot), f_2(\cdot), \ldots, f_k(\cdot))$ be a list of $k$ functions in the RKHS, and we define matrix-like notation $AF(\cdot) := (Af_1(\cdot), \ldots, Af_k(\cdot))$, and $F^\top AF$ is a $k \times k$ matrix, whose $(i, j)$-th element is $\langle f_i, Af_j \rangle_\mathcal{F}$. The outer-product of a function $v \in \mathcal{F}$ defines a linear operator $vv^\top$ such that $(vv^\top)f(\cdot) := \langle v, f \rangle_\mathcal{F} v(\cdot)$, $\forall f \in \mathcal{F}$. Let $V = (v_1(\cdot), \ldots, v_k(\cdot))$ be a list of $k$ functions, then the weighted sum of a set of linear operators can be denoted using matrix-like notation as $V\Sigma_k V^\top := \sum_{i=1}^k \lambda_i v_i v_i^\top$, where $\Sigma_k$ is a diagonal matrix with $\lambda_i$ on the $i$-th entry of the diagonal.

**Eigenfunctions and Kernel PCA** A function $v$ is an eigenfunction of $A$ with the corresponding eigenvalue $\lambda$ if $Av(\cdot) = \lambda v(\cdot)$. Given a set of eigenfunctions $\{v_i\}$ and associated eigenvalues $\{\lambda_i\}$, where $\langle v_i, v_j \rangle_\mathcal{F} = \delta_{ij}$, we can write the eigen-decomposion as $A = V\Sigma_k V^\top + V_\perp \Sigma_\perp V_\perp^\top$, where $V$ is the list of top $k$ eigenfunctions, $\Sigma_k$ is a diagonal matrix with the corresponding eigenvalues, $V_\perp$ is the list of the rest of the eigenfunctions, and $\Sigma_\perp$ is a diagonal matrix with the rest of the eigenvalues.

Kernel PCA aims to identify the the top $k$ subspace $V$. In the finite data case, the empirical co-variance operator is $A = \frac{1}{n} \sum_i k(x_i, \cdot) \otimes k(x_i, \cdot)$. According to the representer theorem, we have $v_i = \sum_{j=1}^n \alpha_i^j k(x_j, \cdot)$, where $\{\alpha_i\}_{i=1}^k \in \mathbb{R}^n$ are weights for the data points. Using $Av(\cdot) = \lambda v(\cdot)$ and the kernel trick, we have $K\alpha_i = \lambda_i \alpha_i$, where $K$ is the $n \times n$ Gram matrix.

**Random feature approximation** The random feature approximation for shift-invariant kernels $k(x, y) = k(x - y)$, e.g., Gaussian RBF kernel, relies on the identity $k(x - y) = \int_{\mathbb{R}^d} e^{i\omega^\top(x-y)} d\mathbb{P}(\omega) = \mathbb{E}[\phi_\omega(x)\phi_\omega(y)]$ since the Fourier transform of a PD function is non-negative, thus can be considered as a (scaled) probability measure [17]. We can therefore approximate the kernel function as an empirical average of samples from the distribution. In other words, $k(x, y) \approx \frac{1}{B} \sum_i \phi_{\omega_i}(x)\phi_{\omega_i}(y)$, where $\{(\omega_i)\}_i^B$ are i.i.d. samples drawn from $\mathbb{P}(\omega)$. For Gaussian RBF kernel, $k(x - x') = \exp(-\|x - x'\|^2/2\sigma^2)$, this yields a Gaussian distribution $\mathbb{P}(\omega) = \exp(-\sigma^2 \|\omega\|^2/2)$. See [17] for more details.

## 4 Algorithm

In this section, we describe an efficient algorithm based on the "doubly stochastic gradients" to scale up kernel PCA. KPCA is essentially an eigenvalue problem in a functional space. Traditional approaches convert it to the dual form, leading to another eigenvalue problem whose size equals the number of training points, which is not scalable. Other approaches solve it in the primal form with stochastic functional gradient descent. However, these algorithms need to store all the training points seen so far. They quickly run into memory issues when working with millions of data points.

We propose to tackle the problem with "doubly stochastic gradients", in which we make two unbiased stochastic approximations. One stochasticity comes from sampling data points as in stochastic gradient descent. Another source of stochasticity is from random features to approximate the kernel.

One technical difficulty in designing doubly stochastic KPCA is an explicit orthogonalization step required in the update rules, which ensures the top $k$ eigenfunctions are orthogonal. This is infeasible for kernel methods on a large dataset since it requires solving an increasingly larger KPCA problem in every iteration. To solve this problem, we formulate the orthogonality constraints into Lagrange multipliers which leads to an Oja-style update rule. The new update enjoys small per iteration complexity and converges to the ground-truth subspace.

We present the algorithm by first deriving the stochastic functional gradient update without random feature approximations, then introducing the doubly stochastic updates. For simplicity of presentation, the following description uses one data point and one random feature at a time, but typically a mini-batch of data points and random features are used in each iteration.

### 4.1 Stochastic functional gradient update

Kernel PCA can be formulated as the following *non-convex* optimization problem

$$\max_{G} \operatorname{tr}\left(G^\top A G\right) \quad \text{s.t.} \, G^\top G = I, \tag{1}$$

where $G := \left(g^1, \ldots, g^k\right)$ and $g^i$ is the $i$-th function.

Gradient descent on the Lagrangian leads to $G_{t+1} = G_t + \eta_t \left(I - G_t G_t^\top\right) A G_t$. Using a stochastic approximation for $A$: $A_t f(\cdot) = f(x_t) k(x_t, \cdot)$, we have $A_t G_t = k(x_t, \cdot) g_t^\top$ and $G_t^\top A_t G_t = g_t g_t^\top$, where $g_t = \left[g_t^1(x_t), \ldots, g_t^k(x_t)\right]^\top$. Therefore, the update rule is

$$G_{t+1} = G_t \left(I - \eta_t g_t g_t^\top\right) + \eta_t k(x_t, \cdot) g_t^\top. \tag{2}$$

This rule can also be derived using stochastic gradient and Oja's rule [15, 16].

### 4.2 Doubly stochastic update

The update rule (2) must store all the data points it has seen so far, which is impractical for large scale datasets. To address this issue, we use the random feature approximation $k(x, \cdot) \approx \phi_{\omega_i}(x) \phi_{\omega_i}(\cdot)$. Denote $H_t$ the function we get at iteration $t$, the update rule becomes

$$H_{t+1} = H_t \left(I - \eta_t h_t h_t^\top\right) + \eta_t \phi_{\omega_t}(x_t) \phi_{\omega_t}(\cdot) h_t^\top, \tag{3}$$

where $h_t$ is the evaluation of $H_t$ at the current data point: $h_t = \left[h_t^1(x_t), \ldots, h_t^k(x_t)\right]^\top$. The specific updates in terms of the coefficients are summarized in Algorithms 1 and 2. Note that in theory new random features are drawn in each iteration, but in practice one can revisit these random features.

### 4.3 Extensions

**Locating individual eigenfunctions** The algorithm only finds the eigen subspace, but not necessarily individual eigenfunctions. A modified version, called Generalized Hebbian Algorithm (GHA) [19] can be used for this purpose: $G_{t+1} = G_t + \eta_t A_t G_t - \eta_t G_t \operatorname{UT}\left[G_t^\top A_t G_t\right]$, where $\operatorname{UT}\left[\cdot\right]$ is an operator that sets the lower triangular parts to zero.

**Latent variable models and kernel SVD** Recently, spectral methods have been proposed to learn latent variable models with provable guarantees [1, 22], in which the key computation is SVD. Our algorithm can be straightforwardly extended to solve kernel SVD, with two simultaneous update rules. The algorithm is summarized in Algorithm 3. See the supplementary for derivation details.

**Kernel CCA and generalized eigenvalue problem** Given two variables $X$ and $Y$, CCA finds two projections such that the correlations between the two projected variables are maximized. It is equivalent to a generalized eigenvalue problem, which can also be solved in our framework. We present the updates for coefficients in Algorithm 4, and derivation details in the supplementary.

**Kernel sliced inverse regression** Kernel sliced inverse regression [10] aims to do sufficient dimension reduction in which the found low dimension representation preserves the statistical correlation with the targets. It also reduces to a generalized eigenvalue problem, and has been shown to find the same subspace as KCCA [10].

## 5 Analysis

In this section, we provide finite time convergence guarantees for our algorithm. As discussed in the previous section, explicit orthogonalization is not scalable for the kernel case, therefore we

| **Algorithm 3: DSGD-KSVD** | **Algorithm 4: DSGD-KCCA** |
|---|---|
| **Require:** $\mathbb{P}(\omega)$, $\phi_\omega(x)$, $k$. | **Require:** $\mathbb{P}(\omega)$, $\phi_\omega(x)$, $k$. |
| **Output:** $\{\alpha_i, \beta_i\}_1^t$ | **Output:** $\{\alpha_i, \beta_i\}_1^t$ |
| 1: **for** $i = 1, \ldots, t$ **do** | 1: **for** $i = 1, \ldots, t$ **do** |
| 2:   Sample $x_i \sim \mathbb{P}(x)$. Sample $y_i \sim \mathbb{P}(y)$. | 2:   Sample $x_i \sim \mathbb{P}(x)$. Sample $y_i \sim \mathbb{P}(y)$. |
| 3:   Sample $\omega_i \sim \mathbb{P}(\omega)$ with seed $i$. | 3:   Sample $\omega_i \sim \mathbb{P}(\omega)$ with seed $i$. |
| 4:   $u_i = \textbf{Evaluate}(x_i, \{\alpha_j\}_{j=1}^{i-1}) \in \mathbb{R}^k$. | 4:   $u_i = \textbf{Evaluate}(x_i, \{\alpha_j\}_{j=1}^{i-1}) \in \mathbb{R}^k$. |
| 5:   $v_i = \textbf{Evaluate}(y_i, \{\beta_j\}_{j=1}^{i-1}) \in \mathbb{R}^k$. | 5:   $v_i = \textbf{Evaluate}(y_i, \{\beta_j\}_{j=1}^{i-1}) \in \mathbb{R}^k$. |
| 6:   $W = u_i v_i^\top + v_i u_i^\top$ | 6:   $W = u_i v_i^\top + v_i u_i^\top$ |
| 7:   $\alpha_i = \eta_i \phi_{\omega_i}(x_i) v_i$. | 7:   $\alpha_i = \eta_i \phi_{\omega_i}(x_i) [v_i - W u_i]$. |
| 8:   $\beta_i = \eta_i \phi_{\omega_i}(y_i) u_i$. | 8:   $\beta_i = \eta_i \phi_{\omega_i}(y_i) [u_i - W v_i]$. |
| 9:   $\alpha_j = \alpha_j - \eta_i W \alpha_j$, for $j = 1, \ldots, i-1$. | 9: **end for** |
| 10:   $\beta_j = \beta_j - \eta_i W \beta_j$, for $j = 1, \ldots, i-1$. | |
| 11: **end for** | |

need to provide guarantees for the updates without orthogonalization. This challenge is even more prominent when using random features, since it introduces additional variance.

Furthermore, our guarantees are w.r.t. the top $k$-dimension subspace. Although the convergence without normalization for a top eigenvector has been established before [15, 16], the subspace case is complicated by the fact that there are $k$ angles between $k$-dimension subspaces, and we need to bound the *largest* angle. To the best of our knowledge, our result is the first finite time convergence result for a subspace *without* explicit orthogonalization.

Note that even though it appears our algorithm is similar to [6] on the surface, the underlying analysis is fundamentally different. In [6], the result only applies to *convex problems* where every local optimum is a global optimum while the problems we consider are highly *non-convex*. As a result, many techniques that [6] builds upon are not applicable.

**Conditions and Assumptions** We will focus on the case when a good initialization $V_0$ is given:

$$V_0^\top V_0 = I, \quad \cos^2 \theta(V, V_0) \geq 1/2. \tag{4}$$

In other words, we analyze the later stage of the convergence, which is typical in the literature (*e.g.*, [21]). The early stage can be analyzed using established techniques (*e.g.*, [3]). In practice, one can achieve a good initialization by solving a small RCA problem [12] with, e.g. thousands, of data points and random features.

Throughout the paper we suppose $|k(x, x')| \leq \kappa, |\phi_\omega(x)| \leq \phi$ and regard $\kappa$ and $\phi$ as constants. Note that this is true for all the kernels and corresponding random features considered. We further regard the eigengap $\lambda_k - \lambda_{k+1}$ as a constant, which is also true for typical applications and datasets.

## 5.1 Update without random features

Our guarantee is on the cosine of the principal angle between the computed subspace and the ground truth eigen subspace (also called potential function): $\cos^2 \theta(V, G_t) = \min_w \frac{\|V^\top G_t w\|^2}{\|G_t w\|^2}$.

Consider the two different update rules, one with explicit orthogonalization and another without

$$F_{t+1} \leftarrow \textbf{orth}(F_t + \eta_t A_t F_t)$$

$$G_{t+1} \leftarrow G_t + \eta_t \left(I - G_t G_t^\top\right) A_t G_t$$

where $A_t$ is the empirical covariance of a mini-batch. Our final guarantee for $G_t$ is the following.

**Theorem 1** *Assume (4) and suppose the mini-batch sizes satisfy that for any $1 \leq i \leq t$, $\|A - A_i\| < (\lambda_k - \lambda_{k+1})/8$. There exist step sizes $\eta_i = O(1/i)$ such that*

$$1 - \cos^2 \theta(V, G_t) = O(1/t).$$

The convergence rate $O(1/t)$ is in the same order as that of computing only the top eigenvector in linear PCA [3]. The bound requires the mini-batch size is large enough so that the spectral norm of $A$ is approximated up to the order of the eigengap. This is because the increase of the potential is in the order of the eigengap. Similar terms appear in the analysis of the noisy power method [7] which, however, requires orthogonalization and is not suitable for the kernel case. We do not specify the

mini-batch size, but by assuming suitable data distributions, it is possible to obtain explicit bounds; see for example [23, 4].

**Proof sketch** We first prove the guarantee for the orthogonalized subspace $F_t$ which is more convenient to analyze, and then show that the updates for $F_t$ and $G_t$ are first order equivalent so $G_t$ enjoys the same guarantee. To do so, we will require lemma 2 and 3 below

**Lemma 2** $1 - \cos^2 \theta(V, F_t) = O(1/t)$.

Let $c_t^2$ denote $\cos^2 \theta(V, F_t)$, then a key step in proving the lemma is to show the following recurrence

$$c_{t+1}^2 \geq c_t^2 (1 + 2\eta_t(\lambda_k - \lambda_{k+1} - 2\|A - A_t\|)(1 - c_t^2)) - O(\eta_t^2). \tag{5}$$

We will need the mini-batch size large enough so that $2\|A - A_t\|$ is smaller than the eigen-gap.

Another key element in the proof of the theorem is the first order equivalence of the two update rules. To show this, we introduce $F(G_t) \leftarrow \mathbf{orth}(G_t + \eta_t A_t G_t)$ to denote the subspace by applying the update rule of $F_t$ on $G_t$. We show that the potentials of $G_{t+1}$ and $F(G_t)$ are close:

**Lemma 3** $\cos^2 \theta(V, G_{t+1}) = \cos^2 \theta(V, F(G_t)) \pm O(\eta_t^2)$.

The lemma means that applying the two update rules to the same input will result in two subspaces with similar potentials. Then by (5), we have $1 - \cos^2 \theta(V, G_t) = O(1/t)$ which leads to our theorem. The proof of Lemma 3 is based on the observation that $\cos^2 \theta(V, X) = \lambda_{\min}(V^\top X(X^\top X)^{-1} X^\top V)$. Comparing the Taylor expansions w.r.t. $\eta_t$ for $X = G_{t+1}$ and $X = F(G_t)$ leads to the lemma.

## 5.2 Doubly stochastic update

The $H_t$ computed in the doubly stochastic update is no longer in the RKHS so the principal angle is not well defined. Instead, we will compare the evaluation of functions from $H_t$ and the true principal subspace $V$ respectively on a point $x$. Formally, we show that for any function $v \in V$ with unit norm $\|v\|_{\mathcal{F}} = 1$, there exists a function $h$ in $H_t$ such that for any $x$, $\text{err} := |v(x) - h(x)|^2$ is small with high probability.

To do so, we need to introduce a companion update rule: $\tilde{G}_{t+1} \leftarrow \tilde{G}_t + \eta_t k(x_t, \cdot) h_t^\top - \eta_t \tilde{G}_t h_t h_t^\top$ resulting in function in the RKHS, but the update makes use of function values from $h_t \in H_t$ which is outside the RKHS. Let $w = \tilde{G}^\top v$ be the coefficients of $v$ projected onto $\tilde{G}$, $h = H_t w$, and $z = \tilde{G}_t w$. Then the error can be decomposed as

$$|v(x) - h(x)|^2 = |v(x) - z(x) + z(x) - h(x)|^2 \leq 2|v(x) - z(x)|^2 + 2|z(x) - h(x)|^2$$
$$\leq \underbrace{2\kappa^2 \|v - z\|_{\mathcal{F}}^2}_{\text{(I: Lemma 5)}} + \underbrace{2|z(x) - h(x)|^2}_{\text{(II: Lemma 6)}}. \tag{6}$$

By definition, $\|v - z\|_{\mathcal{F}}^2 = \|v\|_{\mathcal{F}}^2 - \|z\|_{\mathcal{F}}^2 \leq 1 - \cos^2 \theta(V, \tilde{G}_t)$, so the first error term can be bounded by the guarantee on $\tilde{G}_t$, which can be obtained by similar arguments in Theorem 1. For the second term, note that $\tilde{G}_t$ is defined in such a way that the difference between $z(x)$ and $h(x)$ is a martingale, which can be bounded by careful analysis.

**Theorem 4** *Assume (4) and suppose the mini-batch sizes satisfy that for any $1 \leq i \leq t$, $\|A - A_i\| < (\lambda_k - \lambda_{k+1})/8$ and are of order $\Omega(\ln \frac{t}{\delta})$. There exist step sizes $\eta_i = O(1/i)$, such that the following holds. If $\Omega(1) = \lambda_k(\tilde{G}_i^\top \tilde{G}_i) \leq \lambda_1(\tilde{G}_i^\top \tilde{G}_i) = O(1)$ for all $1 \leq i \leq t$, then for any $x$ and any function $v$ in the span of $V$ with unit norm $\|v\|_{\mathcal{F}} = 1$, we have that with probability at least $1 - \delta$, there exists $h$ in the span of $H_t$ satisfying $|v(x) - h(x)|^2 = O\left(\frac{1}{t} \ln \frac{t}{\delta}\right)$.*

The point-wise error scales as $\tilde{O}(1/t)$ with the step $t$. Besides the condition that $\|A - A_i\|$ is up to the order of the eigengap, we additionally need that the random features approximate the kernel function up to constant accuracy on all the data points up to time $t$, which eventually leads to $\Omega(\ln \frac{t}{\delta})$ mini-batch sizes. Finally, we need $\tilde{G}_i^\top \tilde{G}_i$ to be roughly isotropic, *i.e.*, $\tilde{G}_i$ is roughly orthonormal. Intuitively, this should be true for the following reasons: $\tilde{G}_0$ is orthonormal; the update for $\tilde{G}_t$ is close to that for $G_t$, which in turn is close to $F_t$ that are orthonormal.

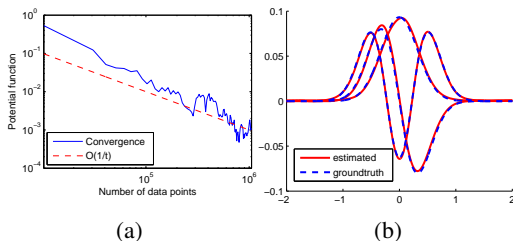
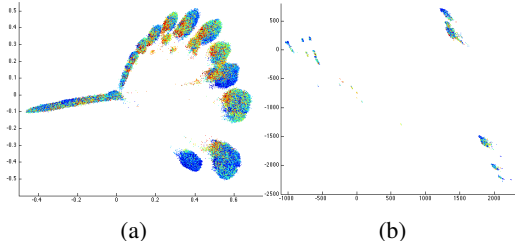

| (a) | (b) | | (a) | (b) |

Figure 1: (a) Convergence of our algorithm on the synthetic dataset. It is on par with the $\tilde{O}(1/t)$ rate denoted by the dashed red line. (b) Recovery of the top three eigenfunctions. Our algorithm (in red) matches the ground-truth (dashed blue).

Figure 2: Visualization of the molecular space dataset by the first two principal components. Bluer dots represent lower PCE values while redder dots are for higher PCE values. (a) Kernel PCA; (b) linear PCA. (Best viewed in color)

**Proof sketch** In order to bound term I in (6), we show that

**Lemma 5** $1 - \cos^2 \theta(V, \tilde{G}_t) = O\left(\frac{1}{t} \ln \frac{t}{\delta}\right)$.

This is proved by following similar arguments to get the recurrence (5), except with an additional error term, which is caused by the fact that the update rule for $\tilde{G}_{t+1}$ is using the evaluation $h_t(x_t)$ rather than $\tilde{g}_t(x_t)$. Bounding this additional term thus relies on bounding the difference between $h_t(x) - \tilde{g}_t(x)$, which is also what we need for bounding term II in (6). For this, we show:

**Lemma 6** *For any $x$ and unit vector $w$, with probability $\geq 1 - \delta$ over $(\mathcal{D}^t, \omega^t)$, $|\tilde{g}_t(x)w - h_t(x)w|^2 = O\left(\frac{1}{t} \ln\left(\frac{t}{\delta}\right)\right)$.*

The key to prove this lemma is that our construction of $\tilde{G}_t$ makes sure that the difference between $\tilde{g}_t(x)w$ and $h_t(x)w$ consists of their difference in each time step. Furthermore, the difference forms a martingale and thus can be bounded by Azuma's inequality. See the supplementary for the details.

## 6 Experiments

**Synthetic dataset with analytical solution** We first verify the convergence rate of DSGD-KPCA on a synthetic dataset with analytical solution of eigenfunctions [24]. If the data follow a Gaussian distribution, and we use a Gaussian kernel, then the eigenfunctions are given by the Hermite polynomials. We generated 1 million such 1-D data points, and ran DSGD-KPCA with a total of 262144 random features. In each iteration, we use a data mini-batch of size 512, and a random feature mini-batch of size 128. After all random features are generated, we revisit and adjust the coefficients of existing random features. The kernel bandwidth is set as the true bandwidth. The step size is scheduled as $\eta_t = \theta_0/(1 + \theta_1 t)$, where $\theta_0$ and $\theta_1$ are two parameters. We use a small $\theta_1 \approx 0.01$ such that in early stages the step size is large enough to arrive at a good initial solution. Figure 1a shows the convergence rate of the proposed algorithm seeking top $k = 3$ subspace. The potential function is the squared sine of the principle angle. We can see the algorithm indeed converges at the rate $O(1/t)$. Figure 1b show the recovered top $k = 3$ eigenfunctions compared with the ground-truth. The solution coincides with one eigenfunction, and deviates only slightly from two others.

**Kernel PCA visualization on molecular space dataset** MolecularSpace dataset contains 2.3 million molecular motifs [6]. We are interested in visualizing the dataset with KPCA. The data are represented by sorted Coulomb matrices of size $75 \times 75$ [14]. Each molecule also has an attribute called power conversion efficiency (PCE). We use a Gaussian kernel with bandwidth chosen by the "median trick". We ran kernel PCA with a total of 16384 random features, with a feature mini-batch size of 512, and data mini-batch size of 1024. We ran 4000 iterations with step size $\eta_t = 1/(1 + 0.001 * t)$. Figure 2 presents visualization by projecting the data onto the top two principle components. Compared with linear PCA, KPCA shrinks the distances between the clusters and brings out the important structures in the dataset. We can also see higher PCE values tend to lie towards the center of the ring structure.

**Nonparametric Latent Variable Model** We learn latent variable models with DSGD-KSVD using one million data points [22], achieving higher quality solutions compared with two other approaches. The dataset consists of two latent components, one is a Gaussian distribution and the other a Gamma distribution with shape parameter $\alpha = 1.2$. DSGD-KSVD uses a total of 8192 random features, and uses a feature mini-batch of size 256 and a data mini-batch of size 512. We compare with 1) random

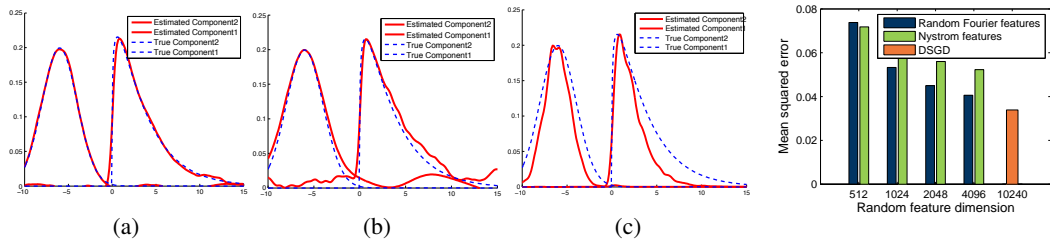

Figure 3: Recovered latent components: (a) DSGD-KSVD, (b) 2048 random features, (c) 2048 Nystrom features.

Figure 4: Comparison on KUKA dataset.

Fourier features, and 2) random Nystrom features, both of fixed 2048 functions [12]. Figures 3 shows the learned conditional distributions for each component. We can see DSGD-KSVD achieves almost perfect recovery, while Fourier and Nystrom random feature methods either confuse high density areas or incorrectly estimate the spread of conditional distributions.

**KCCA MNIST8M** We compare our algorithm on MNIST8M in the KCCA task, which consists of 8.1 million digits and their transformations. We divide each image into the left and right halves, and learn their correlations. The evaluation criteria is the total correlations on the top $k = 50$ canonical correlation directions measured on a separate test set of size 10000. We compare with 1) random Fourier and 2) random Nystrom features on both total correlation and running time. Our algorithm uses a total of 20480 features, with feature mini-batches of size 2048 and data mini-batches of size 1024, and with 3000 iterations. The kernel bandwidth is set using the "median trick" and is the same for all methods. All algorithms are run 5 times, and the mean is reported. The results are presented in Table 1. Our algorithm achieves the best test-set correlations in comparable run time with random Fourier features. This is especially significant for random Fourier features, since the run time would increase by almost four times if double the number of features were used. In addition, Nystrom features generally achieve better results than Fourier features since they are data dependent. We can also see that for large datasets, it is important to use more random features for better performance.

Table 1: KCCA results on MNIST 8M (top 50 largest correlations)

| # of feat | Random features | | Nystrom features | |
|---|---|---|---|---|
| | corrs. | minutes | corrs. | minutes |
| 256 | 25.2 | 3.2 | 30.4 | 3.0 |
| 512 | 30.7 | 7.0 | 35.3 | 5.1 |
| 1024 | 35.3 | 13.9 | 38.0 | 10.1 |
| 2048 | 38.8 | 54.3 | 41.1 | 27.0 |
| 4096 | 41.5 | 186.7 | 42.7 | 71.0 |

| DSGD-KCCA (20480) | |
|---|---|
| corrs. | minutes |
| **43.5** | 183.2 |
| linear CCA | |
| corrs. | minutes |
| 27.4 | 1.1 |

**Kernel sliced inverse regression on KUKA dataset** We evaluate our algorithm under the setting of kernel sliced inverse regression [10], a way to perform sufficient dimension reduction (SDR) for high dimension regression. After performing SDR, we fit a linear regression model using the projected input data, and evaluate mean squared error (MSE). The dataset records rhythmic motions of a KUKA arm at various speeds, representing realistic settings for robots [13]. We use a variant that contains 2 million data points generated by the SL simulator. The KUKA robot has 7 joints, and the high dimension regression problem is to predict the torques from positions, velocities and accelerations of the joints. The input has 21 dimensions while the output is 7 dimensions. Since there are seven independent joints, we set the reduced dimension to be seven. We randomly select 20% as test set and out of the remaining training set, we randomly choose 5000 as validation set to select step sizes. The total number of random features is 10240, with mini-feature batch and mini-data batch both equal to 1024. We run a total of 2000 iterations using step size $\eta_t = 15/(1+0.001*t)$. Figure 4 shows the regression errors for different methods. The error decreases with more random features, and our algorithm achieves lowest MSE by using 10240 random features. Nystrom features do not perform as well in this setting probably because the spectrum decreases slowly (there are seven independent joints) as Nystrom features are known to work well for fast decreasing spectrum.

**Acknowledge**

The research was supported in part by NSF/NIH BIGDATA 1R01GM108341, ONR N00014-15-1-2340, NSF IIS-1218749, NSF CAREER IIS-1350983, NSF CCF-0832797, CCF-1117309, CCF-1302518, DMS-1317308, Simons Investigator Award, and Simons Collaboration Grant.

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
