[Supplementary Material · scalable-KPCA-supplementary.pdf]

# Scale Up Nonlinear Component Analysis with Doubly Stochastic Gradients: Supplementary Material

This supplementary material is organized as follows. Section 1 reviews notations, the definition of Kernel PCA and the update rules considered. Section 2 provides the sketch of the proof as in the paper. Section 3 provides the proof for the stochastic update rule, and Section 4 provides the proof for the doubly stochastic update rule. Section 5 provides details about different extensions.

## 1  Setting

**Notations**   Given a distribution $\mathbb{P}(x)$, a kernel function $k(x, x')$ with RKHS $\mathcal{F}$, the covariance operator $A : \mathcal{F} \mapsto \mathcal{F}$ is a linear self-adjoint operator defined as

$$Af(\cdot) := \mathbb{E}_x[f(x)\, k(x, \cdot)], \quad \forall f \in \mathcal{F}, \tag{1}$$

and furthermore

$$\langle g, Af \rangle_{\mathcal{F}} = \mathbb{E}_x[f(x)\, g(x)], \quad \forall g \in \mathcal{F}.$$

Let $F = (f_1(\cdot), f_2(\cdot), \ldots, f_k(\cdot))$ be a list of $k$ functions in the RKHS, and we define matrix-like notation

$$AF(\cdot) := (Af_1(\cdot), \ldots, Af_k(\cdot)), \tag{2}$$

and $F^\top AF$ is a $k \times k$ matrix, whose $(i, j)$-th element is $\langle f_i, Af_j \rangle_{\mathcal{F}}$. The outer-product of a function $v \in \mathcal{F}$ defines a linear operator $vv^\top : \mathcal{F} \mapsto \mathcal{F}$ such that

$$(vv^\top)f(\cdot) := \langle v, f \rangle_{\mathcal{F}}\, v(\cdot), \quad \forall f \in \mathcal{F} \tag{3}$$

Let $V = (v_1(\cdot), \ldots, v_k(\cdot))$ be a list of $k$ functions, then the weighted sum of a set of linear operators, $\left\{ v_i v_i^\top \right\}_{i=1}^k$, can be denoted using matrix-like notation as

$$V \Sigma_k V^\top := \sum_{i=1}^k \lambda_i v_i v_i^\top \tag{4}$$

where $\Sigma_k$ is a diagonal matrix with $\lambda_i$ on the $i$-th entry of the diagonal.

**Kernel PCA**   Kernel PCA aims to identify the top $k$ eigenfunctions $V = (v_1(\cdot), \ldots, v_k(\cdot))$ for the covariance operator $A$, where $V$ is also called the top $k$ subspace for $A$.

A function $v$ is an eigenfunction of covariance operator $A$ with the corresponding eigenvalue $\lambda$ if

$$Av(\cdot) = \lambda v(\cdot). \tag{5}$$

Given a set of eigenfunctions $\{v_i\}$ and associated eigenvalues $\{\lambda_i\}$, where $\langle v_i, v_j \rangle_{\mathcal{F}} = \delta_{ij}$. We can denote the eigenvalue of $A$ as

$$A = V \Sigma_k V^\top + V_\perp \Sigma_\perp V_\perp^\top \tag{6}$$

where $V = (v_1(\cdot), \ldots, v_k(\cdot))$ is the top $k$ eigenfunctions of $A$, and $\Sigma_k$ is a diagonal matrix with the corresponding eigenvalues, $V_\perp$ is the collection of the rest of the eigenfunctions, and $\Sigma_\perp$ is a diagonal matrix with the rest of the eigenvalues.

**Update rules** The stochastic update rule is

$$G_{t+1} = G_t + \eta_t \left( I - G_t G_t^\top \right) A_t G_t \tag{7}$$

where $G_t := \left( g_t^1, \ldots, g_t^k \right)$ and $g_t^i$ is the $i$-th function. Denote the evaluation of $G_t$ at the current data point as

$$g_t = \left[ g_t^1(x_t), \ldots, g_t^k(x_t) \right]^\top \in \mathbb{R}^k. \tag{8}$$

Then the update rule can be re-written as

$$G_{t+1} = G_t \left( I - \eta_t g_t g_t^\top \right) + \eta_t k(x_t, \cdot) g_t^\top. \tag{9}$$

The doubly stochastic update rule is

$$H_{t+1} = H_t \left( I - \eta_t h_t h_t^\top \right) + \eta_t \phi_{\omega_t}(x_t) \phi_{\omega_t}(\cdot) h_t^\top, \tag{10}$$

where $h_t$ is the evaluation of $H_t$ at the current data point:

$$h_t = \left[ h_t^1(x_t), \ldots, h_t^k(x_t) \right]^\top \in \mathbb{R}^k. \tag{11}$$

When larger mini-batch sizes are used, the update rule is adjusted accordingly. For example, when using $B_{x,t}$ points $\left\{ x_t^b \right\}$ and $B_{\omega,t}$ features $\left\{ \omega_t^{b'} \right\}$, the update rule for $H_t$ is

$$H_{t+1} \leftarrow H_t + \frac{\eta_t \sum_{b,b'} \left( \phi_{\omega_t^{b'}}(x_t^b) \phi_{\omega_t^{b'}}(\cdot) \left[ h_t^1(x_t^b), \ldots, h_t^k(x_t^b) \right] \right)}{B_{x,t} B_{\omega,t}}$$

$$- \eta_t H_t \left( \frac{1}{B_{x,t}} \sum_b \left[ h_t^i(x_t^b) h_t^j(x_t^b) \right] \right)_{i,j=1}^k.$$

# 2  Analysis Roadmap

In order to analyze the convergence of our doubly stochastic kernel PCA algorithm, we will need to define a few intermediate subspaces. For simplicity of notation, we will assume the mini-batch size for the data points is one.

1. Let $F_t := \left( f_t^1, \ldots, f_t^k \right)$ be the subspace estimated using stochastic gradient and explicit orthogonalization:

$$\tilde{F}_{t+1} \leftarrow F_t + \eta_t A_t F_t \tag{12}$$

$$F_{t+1} \leftarrow \tilde{F}_{t+1} \left( \tilde{F}_{t+1}^\top \tilde{F}_{t+1} \right)^{-1/2}$$

2. Let $G_t := \left( g_t^1, \ldots, g_t^k \right)$ be the subspace estimated using stochastic update rule without orthogonalization:

$$G_{t+1} \leftarrow G_t + \eta_t \left( I - G_t G_t^\top \right) A_t G_t. \tag{13}$$

where $A_t G_t$ and $G_t G_t^\top A_t G_t$ can be equivalently written using the evaluation of the function $\left\{ g_t^i \right\}$ on the current data point, leading to the equivalent rule (9):

$$G_{t+1} \leftarrow G_t \left( I - \eta_t g_t g_t^\top \right) + \eta_t k(x_t, \cdot) g_t^\top.$$

3. Let $\tilde{G}_t := \left( \tilde{g}_t^1, \ldots, \tilde{g}_t^k \right)$ be the subspace estimated using stochastic update rule without orthogonalization, but the evaluation of the function $\{\tilde{g}_t^i\}$ on the current data point is replaced by the evaluation $h_t = \left[ h_t^i(x_t) \right]^\top$:

$$\tilde{G}_{t+1} \leftarrow \tilde{G}_t + \eta_t k(x_t, \cdot) h_t^\top - \eta_t \tilde{G}_t h_t h_t^\top \tag{14}$$

4. Let $H_t := \left( h_t^1, \ldots, h_t^k \right)$ be the subspace estimated using doubly stochastic update rule without orthogonalization, $i.e.$, the update rule (10):

$$H_{t+1} \leftarrow H_t + \eta_t \phi_{\omega_t}(x_t) \phi_{\omega_t}(\cdot) h_t^\top - \eta_t H_t h_t h_t^\top. \tag{15}$$

The relation of these subspaces are summarized in Table 1. Using these notations, we describe a sketch of our analysis in the rest of the section, while the complete proofs are provided in the following sections.

We first consider the subspace $G_t$ estimated using the stochastic update rule, since it is simpler and its proof can provide the bases for analyzing the subspace $H_t$ estimated by the doubly stochastic update rule.

Table 1: Relation between various subspaces.

| Subspace | Evaluation | Orth. | Data Mini-batch | RF Mini-batch |
|----------|------------|-------|-----------------|---------------|
| $V$ | − | − | − | − |
| $F_t$ | $f_t(x)$ | ✓ | ✓ | ✗ |
| $G_t$ | $g_t(x)$ | ✗ | ✓ | ✗ |
| $\tilde{G}_t$ | $\tilde{g}_t(x)$ | ✗ | ✓ | ✗ |
| $H_t$ | $h_t(x)$ | ✗ | ✓ | ✓ |

## 2.1 Stochastic update

Our guarantee is on the cosine of the principal angle between the computed subspace and the ground truth eigen subspace $V$ (also called the potential function), which is a standard criterion for measuring the quality of the subspace:

$$\cos^2 \theta(V, G_t) = \min_w \frac{\left\| V^\top G_t w \right\|^2}{\left\| G_t w \right\|^2}.$$

We will focus on the case when a good initialization $V_0$ is given:

$$V_0^\top V_0 = I, \quad \cos^2 \theta(V, V_0) \geq 1/2. \tag{16}$$

In other words, we analyze the later stage of the convergence, which is typical in the literature ($e.g.$, [5]). The early stage can be analyzed using established techniques ($e.g.$, [1]).

We will also focus on the dependence of the potential function on the step $t$. For this reason, throughout the paper we suppose $|k(x, x')| \leq \kappa, |\phi_\omega(x)| \leq \phi$ and regard $\kappa$ and $\phi$ as constants. Note that this is true for all the kernels and corresponding random features considered. We further regard the eigengap $\lambda_k - \lambda_{k+1}$ as a constant, which is also true for typical applications and datasets. Details can be found in the following sections.

Our final guarantee for $G_t$ is stated in the following.

**Theorem 2.** *Assume (16) and suppose the mini-batch sizes satisfy that for any $1 \leq i \leq t$, $\|A - A_i\| < (\lambda_k - \lambda_{k+1})/8$. There exist step sizes $\eta_i = O(1/i)$ such that*

$$1 - \cos^2 \theta(V, G_t) = O(1/t).$$

The convergence rate $O(1/t)$ is in the same order as that when computing only the top eigenvector in linear PCA [1], though we are not aware of any other convergence rate for computing the top $k$ eigenfunctions in Kernel PCA. The bound requires the mini-batch sizes are large enough so that the spectral norm of $A$ is approximated up to the order of the eigengap. This is due to the fact that approximating $A$ with $A_t$ will result in an error term in the order of $\|A - A_t\|$, while the increase of the potential is in the order of the eigengap. Similar terms appear in the analysis of the noisy power method [3] which, however, requires normalization and is not suitable for the kernel case. We do not specify the mini-batch sizes, but by assuming suitable data distributions, it is possible to obtain explicit bounds; see for example [6, 2].

**Proof sketch** To prove the theorem, we first prove the guarantee for the normalized subspace $F_t$ which is more convenient to analyze, and then show that the update rules for $F_t$ and $G_t$ are first order equivalent so that $G_t$ enjoys the same guarantee.

**Lemma 3.** $1 - \cos^2 \theta(V, F_t) = O(1/t)$.

Let $c_t^2$ denote $\cos^2 \theta(V, F_t)$, then a key step in proving the lemma is to show that

$$c_{t+1}^2 \geq c_t^2 (1 + 2\eta_t(\lambda_k - \lambda_{k+1} - 2\|A - A_t\|)(1 - c_t^2)) - O(\eta_t^2). \tag{17}$$

Therefore, we will need the mini-batch sizes large enough so that $2\|A - A_t\|$ is smaller than the eigen-gap.

Another key element in the proof of the theorem is the first order equivalence of the two update rules. To show this, we need to compare the subspaces obtained by applying the them on the same subspace $G_t$. So we introduce $F(G_t)$ to denote the subspace by applying the update rule of $F_t$ on $G_t$:

$$\tilde{F}(G_t) \leftarrow G_t + \eta_t A_t G_t$$

$$F(G_t) \leftarrow \tilde{F}(G_t) \left[ \tilde{F}(G_t)^\top \tilde{F}(G_t) \right]^{-1/2}$$

We show that the potentials of $G_{t+1}$ and $F(G_t)$ are close:

**Lemma 4.** $\cos^2 \theta(V, G_{t+1}) = \cos^2 \theta(V, F(G_t)) \pm O(\eta_t^2)$.

The lemma means that applying the two update rules to the same input will result in two subspaces with similar potentials. Since $\cos^2 \theta(V, F(G_t))$ enjoys the recurrence in (17), we know that $\cos^2 \theta(V, G_{t+1})$ also enjoys such a recurrence, which then results in $1 - \cos^2 \theta(V, G_t) = O(1/t)$.

The proof of the lemma is based on the observation that

$$\cos^2 \theta(V, X) = \lambda_{\min}(V^\top X (X^\top X)^{-1} X^\top V).$$

The lemma follows by plugging in $X = G_{t+1}$ or $X = F(G_t)$ and comparing their Taylor expansions w.r.t. $\eta_t$.

## 2.2 Doubly stochastic update

For doubly stochastic update rule, the computed $H_t$ is no longer in the RKHS so the principal angle is not well defined. Since the eigenfunction $v$ is usually used for evaluating on points $x$, we will use the following point-wise convergence in our analysis. For any function $v$ in the subspace of $V$ with unit norm $\|v\|_{\mathcal{F}} = 1$, we will find a specially chosen function $h$ in the subspace of $H_t$ such that for any $x$,

$$\text{err} := |v(x) - h(x)|^2$$

is small with high probability. More specifically, the $w$ is chosen to be $\tilde{G}^\top v$, and let $\tilde{g} = \tilde{G}_t w$ and $h = H_t w$. Then the error measure can be decomposed as

$$
\begin{aligned}
&|v(x) - h(x)|^2 \\
&= |v(x) - \tilde{g}(x) + \tilde{g}(x) - h(x)|^2 \\
&\leq 2\,|v(x) - \tilde{g}(x)|^2 + 2\,|\tilde{g}(x) - h(x)|^2 \\
&\leq \underbrace{2\kappa^2\,\|v - \tilde{g}\|_{\mathcal{F}}^2}_{\text{(I: Lemma 6)}} + \underbrace{2\,|\tilde{g}(x) - h(x)|^2}_{\text{(II: Lemma 7)}}.
\end{aligned}
\tag{18}
$$

The distance $\|v - \tilde{g}\|_{\mathcal{F}}$ is closely related to the squared sine of the subspace angle between $V$ and $\tilde{G}_t$. In fact, by definition, $\|v - \tilde{g}\|_{\mathcal{F}}^2 = \|v\|_{\mathcal{F}}^2 - \|\tilde{g}\|_{\mathcal{F}}^2 \leq 1 - \cos^2 \theta(V, \tilde{G}_t)$. Therefore, the first error term can be bounded by the guarantee on $\tilde{G}_t$, which can be obtained by similar arguments as for the stochastic update case. For the second term, note that $\tilde{G}_t$ is defined in such a way that the difference between $\tilde{g}(x) = \tilde{G}_t(x)w$ and $h(x) = H_t(x)w$ is a martingale, which can be bounded by careful analysis.

Overall, we have the following results. Suppose we use random Fourier features; see [4]. Similar bounds hold for other random features, where the batch sizes will depend on the concentration bound of the random features used.

**Theorem 5.** *Assume (16) and suppose the mini-batch sizes satisfy that for any $1 \leq i \leq t$, $\|A - A_i\| < (\lambda_k - \lambda_{k+1})/8$ and are of order $\Omega(\ln \frac{t}{\delta})$. There exist step sizes $\eta_i = O(1/i)$, such that the following holds. If $\Omega(1) = \lambda_k(\tilde{G}_i^\top \tilde{G}_i) \leq \lambda_1(\tilde{G}_i^\top \tilde{G}_i) = O(1)$ for all $1 \leq i \leq t$, then for any $x$ and any function $v$ in the span of $V$ with unit norm $\|v\|_{\mathcal{F}} = 1$, we have that with probability $\geq 1 - \delta$, there exists $h$ in the span of $H_t$ satisfying*

$$
|v(x) - h(x)|^2 = O\left(\frac{1}{t}\ln\frac{t}{\delta}\right).
$$

The point-wise error scales as $\tilde{O}(1/t)$ with the step $t$, which is in similar order as that for the stochastic update rule. Again, we require the spectral norm of $A$ to be estimated up to the order of the eigengap, for the same reason as before. We additionally need that the random features approximate the kernel function up to constant accuracy on all the data points up to time $t$, since the evaluation of the kernel function on these points are used in the update. This eventually leads to $\Omega(\ln \frac{t}{\delta})$ mini-batch sizes. Finally, we need $\tilde{G}_i^\top \tilde{G}_i$ to be roughly isotropic, *i.e.*, $\tilde{G}_i$ is roughly orthonormal. Intuitively, this should be true for the following reasons: $\tilde{G}_0$ is orthonormal; the update for $\tilde{G}_t$ is close to that for $G_t$, which in turn is close to $F_t$ that are orthonormal.

**Proof sketch** The analysis is carried out by bounding each term in (18) separately. As discussed above, in order to bound term I, we need a bound on the squared cosine of the subspace angle between $V$ and $\tilde{G}_t$.

**Lemma 6.** $1 - \cos^2 \theta(V, \tilde{G}_t) = O\left(\frac{1}{t}\ln\frac{t}{\delta}\right)$.

To prove this lemma, we follow the argument for Theorem 2 and get the recurrence as shown in (17), except with an additional error term, which is caused by the fact that the update rule for $\tilde{G}_{t+1}$ is using the evaluation $h_t(x_t)$ rather than $\tilde{g}_t(x_t)$. Bounding this additional term thus relies on bounding the difference between $h_t(x) - \tilde{g}_t(x)$, which is also what we need for bounding term II in (18). For this purpose, we show the following bound:

**Lemma 7.** *For any $x$ and unit vector $w$, with probability $\geq 1 - \delta$ over $(\mathcal{D}^t, \omega^t)$, $|\tilde{g}_t(x)w - h_t(x)w|^2 = O\left(\frac{1}{t}\ln\left(\frac{t}{\delta}\right)\right)$.*

The key to prove this lemma is that our construction of $\tilde{G}_t$ makes sure that the difference between $\tilde{g}_t(x)w$ and $h_t(x)w$ consists of their difference in each time step. Furthermore, the difference in each time step conditioned on previous history has mean 0. In other words, the difference forms a martingale and

thus can be bounded by Azuma's inequality. The resulting bound depends on the mini-batch sizes, the step sizes $\eta_i$, and the evaluations $h_i(x_i)$ used in the update rules. We then judiciously choose the parameters and simplify it to the bound in the lemma. The complication of the proof is mostly due to the interweaving of the parameter values; see the following sections for the details.

# 3 Stochastic Update

To prove the convergence of the stochastic update rule, we first prove the convergence of the normalized version $F_t$, and then we establish the first-order equivalence of the potential functions of the two update rules for $F_t$ and $G_t$. Since the final recurrence result does not depend on higher order terms, this first-order equivalence establishes the convergence of the stochastic update rule without normalization.

## 3.1 Stochastic update with normalization

We consider the potential function $1 - \cos^2 \theta (V, F_t)$ and prove a recurrence for it. We first show this for the simpler case where at each step we use the expected operator $A$ in the update rule, and then show this for the general case where $A_t$ can be different from $A$.

### 3.1.1 Update rule with expected operator

The following lemma states the recurrence for the update rule which replace $A_t$ in the stochastic update rule with the expected operator $A = \mathbb{E} A_t$:

$$\tilde{F}_{t+1} \leftarrow F_t + \eta_t A F_t \tag{19}$$

$$F_{t+1} \leftarrow \tilde{F}_{t+1} \left( \tilde{F}_{t+1}^\top \tilde{F}_{t+1} \right)^{-1/2}$$

**Lemma 8.** *Let the sequence $\{F_i\}_i$ be obtained from the update rule (19), then*

$$1 - \cos^2 \theta (V, F_{t+1}) \leq \left[ 1 - \cos^2 \theta (V, F_t) \right] \left[ 1 - 2\eta_t (\lambda_k - \lambda_{k+1}) \cos^2 \theta (V, F_t) \right] + \beta_t,$$

*where $\beta_t = 5\eta_t^2 B^2 + 3\eta_t^3 B^3$ and $\lambda_k$ and $\lambda_{k+1}$ are the top $k$ and $k+1$-th eigenvalues of $A$.*

*Proof.* First note that the cosine of subspace angle does not change under linear combination of the basis

$$\cos^2 \theta (V, F_{t+1}) = \min_{w'} \frac{\left\| V^\top F_{t+1} w' \right\|^2}{\left\| F_{t+1} w' \right\|^2} = \min_{w'} \frac{\left\| V^\top \tilde{F}_{t+1} \left( \tilde{F}_{t+1}^\top \tilde{F}_{t+1} \right)^{-1/2} w' \right\|^2}{\left\| \tilde{F}_{t+1} \left( \tilde{F}_{t+1}^\top \tilde{F}_{t+1} \right)^{-1/2} w' \right\|^2} = \min_{w} \frac{\left\| V^\top \tilde{F}_{t+1} w \right\|^2}{\left\| \tilde{F}_{t+1} w \right\|^2} \tag{20}$$

The update rule gives us

$$\left\| V^\top \tilde{F}_{t+1} w \right\|^2 \geq \left\| V^\top F_t w \right\|^2 + 2\eta_t \left\langle V^\top F_t w, V^\top A F_t w \right\rangle \tag{21}$$

$$\left\| \tilde{F}_{t+1} w \right\|^2 \leq \left\| F_t w \right\|^2 + 2\eta_t \left\langle F_t w, A F_t w \right\rangle + B \left\| F_t w \right\|^2 \eta_t^2 \tag{22}$$

Let $\hat{w} = w/\|F_t w\|$, $u = F_t \hat{w}$, so $\|u\| = 1$. Denote $c = \|V^\top u\|$ and $s = \|V_\perp^\top u\|$. According to the definition, we have $c \geq \cos \theta_k (V, F_t)$. Keep expanding the update rule leads to

$$\frac{\left\|V^\top \tilde{F}_{t+1} w\right\|^2}{\left\|\tilde{F}_{t+1} w\right\|^2} \geq \frac{\|V^\top F_t w\|^2 + 2\eta_t \left\langle V^\top F_t w, V^\top A F_t w\right\rangle}{\|F_t w\|^2 + 2\eta_t \left\langle F_t w, A F_t w\right\rangle + B \|F_t w\|^2 \eta_t^2} \tag{23}$$

$$= \frac{\|V^\top u\|^2 + 2\eta_t \left\langle V^\top u, V^\top A u\right\rangle}{1 + 2\eta_t \left\langle u, A u\right\rangle + B \eta_t^2}$$

$$\geq \left\{\|V^\top u\|^2 + 2\eta_t \left\langle V^\top u, V^\top A u\right\rangle\right\} \left\{1 - 2\eta_t \left\langle u, A u\right\rangle - B \eta_t^2\right\}$$

$$\geq \|V^\top u\|^2 + 2\eta_t \left\langle V^\top u, V^\top A u\right\rangle - 2\eta_t \|V^\top u\|^2 \left\langle u, A u\right\rangle$$
$$- 5\eta_t^2 B^2 - 2\eta_t^3 B^3$$

$$= c^2 + 2\eta_t \left\{u^\top V V^\top A u - c^2 u^\top A u\right\} - \beta_t$$

$$= c^2 + 2\eta_t u^\top \left(V V^\top - c^2 I\right) A u - \beta_t$$

$$= c^2 + 2\eta_t u^\top \left(s^2 V V^\top - c^2 V_\perp V_\perp^\top\right) A u - \beta_t.$$

Recall that $A = V \Lambda_k V^\top + V_\perp \Lambda_{k+1} V_\perp^\top$. Then

$$u^\top \left(s^2 V V^\top - c^2 V_\perp V_\perp^\top\right) A u = s^2 u^\top V \Lambda_k V^\top u - c^2 u^\top V_\perp \Lambda_{k+1} V_\perp^\top u \tag{24}$$
$$\geq \lambda_k s^2 c^2 - \lambda_{k+1} c^2 s^2 = s^2 c^2 \left(\lambda_k - \lambda_{k+1}\right)$$

The recurrence is therefore

$$\cos^2 \theta (V, F_{t+1}) \geq c^2 + 2\eta_t s^2 c^2 \left(\lambda_k - \lambda_{k+1}\right) - \beta_t \tag{25}$$
$$= c^2 \left(1 + 2\eta_t \left(\lambda_k - \lambda_{k+1}\right) \left(1 - c^2\right)\right) - \beta_t.$$

The first term is a quadratic function of $c^2$:

$$x \left(1 + a \left(1 - x\right)\right) \tag{26}$$

where $x := c^2$ and $a = 2\eta_t \left(\lambda_k - \lambda_{k+1}\right)$. It has two roots at $0$ and $1 + \frac{1}{a}$. Therefore, if $\frac{1}{2} + \frac{1}{2a} \geq 1$, it is a monotonic increasing function in the interval of $[0, 1]$.

Thus, if $\eta_t \leq \frac{1}{4(\lambda_k - \lambda_{k+1})}$, which holds for all $t$ large enough, we have

$$\cos^2 \theta (V, F_{t+1}) \geq \cos^2 \theta (V, F_t) \left(1 + 2\eta_t \left(\lambda_k - \lambda_{k+1}\right) \left(1 - \cos^2 \theta (V, F_t)\right)\right) - \beta_t \tag{27}$$

which leads to the lemma. $\qquad\square$

### 3.1.2 Using different operators in different iterations

Now consider the case of stochastic update rule (12) where we use a mini-batch to approximate the expectation in each iteration.

**Lemma 9.** *Let the sequence $\{F_i\}_i$ be obtained from the update rule (12), then*

$$1 - \cos^2 \theta (V, F_{t+1}) \leq \left[1 - \cos^2 \theta (V, F_t)\right] \left[1 - 2\eta_t \left(\lambda_k - \lambda_{k+1} - \|A_t - A\|\right) \cos^2 \theta (V, F_{t+1})\right] + \beta_t,$$

*where $\beta_t = 5\eta_t^2 B^2 + 3\eta_t^3 B^3$ and $\lambda_k$ and $\lambda_{k+1}$ are the top $k$ and $k+1$-th eigenvalues of $A$.*

*Proof.* The effect of the stochastic update is an additional term in the recurrence

$$\cos^2 \theta \left( V, F_{t+1} \right) \geq c^2 + 2\eta_t u^\top \left( s^2 V V^\top - c^2 V_\perp V_\perp^\top \right) A u + Z_t - \beta_t \tag{28}$$

where

$$Z_t = 2\eta_t u^\top \left( s^2 V V^\top - c^2 V_\perp V_\perp^\top \right) \left( A_t - A \right) u. \tag{29}$$

The effect of the noise can be bounded, i.e.

$$\begin{aligned} Z_t &= 2\eta_t s^2 u^\top V V^\top \left( A_t - A \right) u - 2\eta_t c^2 u^\top V_\perp V_\perp^\top \left( A_t - A \right) u \\ &= 2\eta_t s^2 u^\top \left( V V^\top + l_1 I \right) \left( A_t - A \right) u - 2\eta_t c^2 u^\top \left( V_\perp V_\perp^\top + l_2 I \right) \left( A_t - A \right) u, \end{aligned} \tag{30}$$

where $s^2 l_1 = c^2 l_2$ are positive numbers such that $V V^\top + l_1 I$ and $V_\perp V_\perp^\top + l_2 I$ are positive-definite.

The generalized Rayleigh quotient leads to the inequality

$$\begin{aligned} \left| u^\top \left( V V^\top + l_1 I \right) \left( A_t - A \right) u \right| &\leq \lambda u^\top \left( V V^\top + l_1 I \right) u \\ &\leq \lambda \left( c^2 + l_1 \right) \end{aligned} \tag{31}$$

where $\lambda$ is the largest generalized eigen-value that satisfies

$$\left( V V^\top + l_1 I \right) \left( A_t - A \right) x = \lambda \left( V V^\top + l_1 I \right) x. \tag{32}$$

Since $V V^\top + l_1 I$ is positive definite, we have $\lambda = \| A_t - A \|$.

Similarly, we have

$$\left| u^\top \left( V_\perp V_\perp^\top + l_2 I \right) \left( A_t - A \right) u \right| \leq \| A_t - A \| \left( s^2 + l_2 \right). \tag{33}$$

The noise term is thus bounded by

$$Z_t \geq -2\eta_t s^2 \| A_t - A \| \left( c^2 + l_1 \right) - 2\eta_t c^2 \| A_t - A \| \left( s^2 + l_2 \right). \tag{34}$$

Note that $l_1$ and $l_2$ can be infinitely small positive so we can ignore them.

Therefore, the recurrence is

$$\begin{aligned} \cos^2 \theta \left( V, F_{t+1} \right) &\geq c^2 + 2\eta_t s^2 c^2 \left( \lambda_k - \lambda_{k+1} \right) - 4\eta_t \| A_t - A \| s^2 c^2 - \beta_t \\ &= c^2 \left( 1 + 2\eta_t \left( \lambda_k - \lambda_{k+1} - 2 \| A_t - A \| \right) \left( 1 - c^2 \right) \right) - \beta_t \end{aligned} \tag{35}$$

which then leads to the lemma. □

In order to get fast convergence, we need to take sufficiently large mini-batches such that the variance of the noise is small enough compared with the eigen-gap.

## 3.2 Stochastic update without normalization

We show that the cosine angles of the two updates are first-order equivalent. Then, since the recurrence is not affected by higher order terms, when the step size is small enough, we can show it also converges in $O(1/t)$.

To show the first order equivalence, we need to compare the subspaces obtained by applying the them on the same subspace $G_t$. So we introduce $F(G_t)$ to denote the subspace by applying the update rule of $F_t$ on $G_t$:

$$\tilde{F}(G_t) \leftarrow G_t + \eta_t A_t G_t \tag{36}$$

$$F(G_t) \leftarrow \tilde{F}(G_t) \left[ \tilde{F}(G_t)^\top \tilde{F}(G_t) \right]^{-1/2} \tag{37}$$

Then the first order equivalence as stated in Lemma 4 follows from the following two lemmas for the normalized update rule (12) and the unnormalized update rule (36), respectively.

**Lemma 10.** $\cos^2\theta\,(V, F(G_t)) = \lambda_{min}\left(M + O(\eta^2)\right)$ *where*

$$M = V^\top P P^\top V + \eta V^\top P P^\top A V + \eta V^\top A P P^\top V - 2\eta V^\top P P^\top A P P^\top V,$$

*where* $PP^\top = G_t\left(G_t^\top G_t\right)^{-1} G_t^\top$, *and* $P$ *is an orthonormal basis for the subspace* $G_t$.

*Proof.* For simplicity, let $G$ denote $G_t$, and let $A$ denote $A_t$ in the following. We first have

$$\cos^2\theta\,(V, F(G)) = \lambda_{\min}\left(V^\top F(G)F(G)^\top V\right) \tag{38}$$

$$= \lambda_{\min}\left(F(G)^\top V V^\top F(G)\right) \tag{39}$$

$$= \lambda_{\min}\left\{V^\top (G + \eta_t AG)\left[(G + \eta_t AG)^\top (G + \eta_t AG)\right]^{-1} (G + \eta_t AG)^\top V\right\}. \tag{40}$$

Note that (39) is due to the fact that

$$\lambda_{\min}\left(F(G)^\top V V^\top F(G)\right) = \min_w \frac{w^\top F(G)^\top V V^\top F(G)w}{w^\top w}$$

$$= \min_w \frac{w^\top R^{-1}(G + \eta_t AG)^\top V V^\top (G + \eta_t AG) R^{-1}w}{w^\top w}$$

$$= \min_z \frac{z^\top (G + \eta_t AG)^\top V V^\top (G + \eta_t AG)z}{z^\top R^2 z}$$

$$= \min_z \frac{z^\top (G + \eta_t AG)^\top V V^\top (G + \eta_t AG)z}{z^\top (G + \eta_t AG)^\top (G + \eta_t AG)z}$$

$$= \min_z \frac{\left\|V^\top (G + \eta_t AG)z\right\|^2}{\left\|(G + \eta_t AG)z\right\|^2}$$

where $R = \left[(G + \eta_t AG)^\top (G + \eta_t AG)\right]^{1/2}$.

Now turn back to (40). Expand the matrix-valued function

$$\phi(\eta) = \left[(G + \eta AG)^\top (G + \eta AG)\right]^{-1} \tag{41}$$

$$= \phi(0) + \phi'(0)\eta + O(\eta^2).$$

$$\phi'(0) = -2\left(G^\top G\right)^{-1} G^\top A G\left(G^\top G\right)^{-1}. \tag{42}$$

So,

$$\phi(\eta) = \left(G^\top G\right)^{-1} - 2\eta\left(G^\top G\right)^{-1} G^\top A G\left(G^\top G\right)^{-1} + O(\eta^2). \tag{43}$$

Therefore,

$$V^\top (G + \eta_t AG)\left[(G + \eta_t AG)^\top (G + \eta_t AG)\right]^{-1} (G + \eta_t AG)^\top V \tag{44}$$

$$= \left(V^\top G + \eta_t V^\top AG\right)\left[\left(G^\top G\right)^{-1} - 2\eta\left(G^\top G\right)^{-1} G^\top A G\left(G^\top G\right)^{-1} + O(\eta^2)\right]\left(G^\top V + \eta_t G^\top AV\right)$$

$$= V^\top G\left(G^\top G\right)^{-1} G^\top V + \eta V^\top G\left(G^\top G\right)^{-1} G^\top AV + \eta V^\top AG\left(G^\top G\right)^{-1} G^\top V$$

$$- 2\eta V^\top G\left(G^\top G\right)^{-1} G^\top A G\left(G^\top G\right)^{-1} G^\top V + O(\eta^2)$$

$$= V^\top P P^\top V + \eta V^\top P P^\top AV + \eta V^\top A P P^\top V - 2\eta V^\top P P^\top A P P^\top V + O(\eta^2),$$

where $PP^\top = G\left(G^\top G\right)^{-1} G^\top$, and $P$ is an orthonormal basis for the subspace $G$. $\square$

**Lemma 11.** $\cos^2\theta\left(V, G_{t+1}\right) = \lambda_{min}\left(M\right)$ *where* $M$ *is as defined in Lemma 10.*

*Proof.* For simplicity, let $G$ denote $G_t$ and let $A$ denote $A_t$. Then $\cos^2\theta\left(V, G_{t+1}\right) = \lambda_{\min}\left(N\right)$, where

$$N = V^\top G_{t+1}\left[G_{t+1}^\top G_{t+1}\right]^{-1} G_{t+1}^\top V \ \ \text{with} \ \ G_{t+1} = G + \eta\left(I - GG^\top\right)AG.$$

Now it suffices to show $N = M$. Consider

$$\phi(\eta) = \left[\left(G + \eta\left(I - GG^\top\right)AG\right)^\top \left(G + \eta\left(I - GG^\top\right)AG\right)\right]^{-1}.$$

Then

$$\phi'(0) = -\left(G^\top G\right)^{-1}\left[G^\top\left(I - GG^\top\right)AG + G^\top A\left(I - GG^\top\right)G\right]\left(G^\top G\right)^{-1}$$

Therefore, $N$ is

$$V^\top\left(G + \eta\left(I - GG^\top\right)AG\right)\left[\left(G + \eta\left(I - GG^\top\right)AG\right)^\top\left(G + \eta\left(I - GG^\top\right)AG\right)\right]^{-1}\left(G + \eta\left(I - GG^\top\right)AG\right)^\top V$$

$$= \left(V^\top G + \eta V^\top\left(I - GG^\top\right)AG\right)\left[\left(G + \eta\left(I - GG^\top\right)AG\right)^\top\left(G + \eta\left(I - GG^\top\right)AG\right)\right]^{-1}\left(G^\top V + \eta G^\top A\left(I - GG^\top\right)V\right)$$

$$= \left(V^\top G + \eta V^\top\left(I - GG^\top\right)AG\right)$$

$$\left[\left(G^\top G\right)^{-1} - \eta\left(G^\top G\right)^{-1}\left[G^\top\left(I - GG^\top\right)AG + G^\top A\left(I - GG^\top\right)G\right]\left(G^\top G\right)^{-1}\right]\left(G^\top V + \eta G^\top A\left(I - GG^\top\right)V\right)$$

$$= V^\top G\left(G^\top G\right)^{-1}G^\top V + \eta V^\top G\left(G^\top G\right)^{-1}G^\top A\left(I - GG^\top\right)V + \eta V^\top\left(I - GG^\top\right)AG\left(G^\top G\right)^{-1}G^\top V$$

$$- \eta V^\top G\left(G^\top G\right)^{-1}\left[G^\top\left(I - GG^\top\right)AG + G^\top A\left(I - GG^\top\right)G\right]\left(G^\top G\right)^{-1}G^\top V$$

$$= V^\top PP^\top V + \eta V^\top PP^\top A\left(I - GG^\top\right)V + \eta V^\top\left(I - GG^\top\right)APP^\top V$$

$$- \eta V^\top PP^\top\left(I - GG^\top\right)APP^\top V - \eta V^\top PP^\top A\left(I - GG^\top\right)PP^\top V$$

$$= V^\top PP^\top V + \eta V^\top PP^\top AV + \eta V^\top APP^\top V - 2\eta V^\top PP^\top APP^\top V$$

$$- \eta V^\top PP^\top AGG^\top V - \eta V^\top GG^\top APP^\top V + \eta V^\top PP^\top GG^\top APP^\top V + \eta V^\top PP^\top AGG^\top PP^\top V$$

$$= V^\top PP^\top V + \eta V^\top PP^\top AV + \eta V^\top APP^\top V - 2\eta V^\top PP^\top APP^\top V$$

which completes the proof. $\qquad\square$

## 4  Doubly Stochastic Update

In this section, we consider the doubly stochastic update rule. Suppose in step $t$, we use a mini-batch consisting of $B_{x,t}$ random data points $x_t^r (1 \leq r \leq B_{x,t})$ and $B_{\omega,t}$ random features $\omega_t^s (1 \leq s \leq B_{\omega,t})$. Then the update rule is

$$H_{t+1} = H_t + \eta_t \mathbb{E}_t\left[\phi_{\omega_t}(x_t)\phi_{\omega_t}(\cdot)h_t(x_t)\right] - \eta_t H_t \mathbb{E}_t\left[h_t(x_t)^\top h_t(x_t)\right] \tag{45}$$

$$= H_t(I - \eta_t \mathbb{E}_t\left[h_t(x_t)^\top h_t(x_t)\right]) + \eta_t \mathbb{E}_t\left[\phi_{\omega_t}(x_t)\phi_{\omega_t}(\cdot)h_t(x_t)\right] \tag{46}$$

where for any function $f(x,\omega)$, $\mathbb{E}_t f(x_t, \omega)$ denotes $\sum_{r=1}^{B_{x,t}}\sum_{s=1}^{B_{\omega,t}} f(x_t^r, \omega_t^s)/(B_{x,t}B_{\omega,t})$. As before, we assume $H_0 = F_0$ is a good initialization, *i.e.*, $F_0^\top F_0 = I$ and $\cos^2\theta(F_0, V) \geq 1/2$. Note that $H_t = [h_t^1(\cdot), \ldots, h_t^k(\cdot)]$, while $h_t(x_t)$ is its evaluation at $x_t$, *i.e.*, $h_t(x_t)$ is a row vector $[h_t^1(x_t), \ldots, h_t^k(x_t)]$.

We introduce the following intermediate function for analysis:

$$\tilde{G}_{t+1} = \tilde{G}_t + \eta_t \mathbb{E}_t\left[k(x_t, \cdot)h_t(x_t)\right] - \eta_t \tilde{G}_t \mathbb{E}_t\left[h_t(x_t)^\top h_t(x_t)\right] \tag{47}$$

$$= \tilde{G}_t(I - \eta_t \mathbb{E}_t\left[h_t(x_t)^\top h_t(x_t)\right]) + \eta_t \mathbb{E}_t\left[k(x_t, \cdot)h_t(x_t)\right]. \tag{48}$$

Again, $\tilde{G}_0 = F_0$.

The analysis follows our intuition: we first bound the difference between $H_t$ and $\tilde{G}_t$ by a martingale argument, and then bound the difference between $\tilde{G}_t$ and $V$. For the second step we can apply the previous argument. Note that $\tilde{G}_t$ is different from $F_t$ since $A_t F_t = k(x_t, \cdot) F_t(x_t)$ is now replaced by $k(x_t, \cdot) h_t(x_t)$, so we need to adjust our previous analysis.

Suppose we use random Fourier features for points in $\mathbb{R}^d$; see [4]. Then we have

**Theorem 5.** *Suppose the mini-batch sizes satisfy that for any $1 \leq i < t$, $\|A - A_i\| < (\lambda_k - \lambda_{k+1})/8$ and $B_{x,i} = \Omega(\ln \frac{t}{\delta})$. There exist step sizes $\eta_i = O(1/i)$, such that the following holds. If $\Omega(1) = \lambda_k(\tilde{G}_i^\top \tilde{G}_i) \leq \lambda_1(\tilde{G}_i^\top \tilde{G}_i) = O(1)$ for all $1 \leq i \leq t$, then for any $x$ and any function $v$ in the span of $V$ with unit norm $\|v\|_{\mathcal{F}} = 1$, we have that with probability $\geq 1 - \delta$, there exists $h$ in the span of $H_t$ satisfying*

$$|v(x) - h(x)|^2 = O\left(\frac{1}{t} \ln \frac{t}{\delta}\right).$$

*Proof.* Let $w = \tilde{G}_t^\top v$, $z = \tilde{G}_t w$, and $h = H_t w$.

$$
\begin{aligned}
|v(x) - h(x)|^2 &= |v(x) - z(x) + z(x) - h(x)|^2 \\
&\leq 2|v(x) - z(x)|^2 + 2|z(x) - h(x)|^2 \\
&\leq 2\|v - z\|_{\mathcal{F}}^2 \|k(x, \cdot)\|_{\mathcal{F}}^2 + 2|z(x) - h(x)|^2 \\
&\leq 2\kappa^2 \|v - z\|_{\mathcal{F}}^2 + |z(x) - h(x)|^2.
\end{aligned}
$$

Roughly speaking, the difference between $v$ and $z$ is the error due to random data points and can be bounded by Lemma 15, while the difference between $z(x)$ and $h(x)$ is the error due to random features and can be bounded by Lemma 13(2). More precisely, since $z$ is the projection of $v$ on the span of $\tilde{G}_t$,

$$\|v - z\|_{\mathcal{F}}^2 = \|v\|_{\mathcal{F}}^2 - \|z\|_{\mathcal{F}}^2 \leq 1 - \cos^2 \theta(\tilde{G}_t, V) = O\left(\frac{1}{t} \ln \frac{t}{\delta}\right)$$

where the last step is by Lemma 15. Also, since $\|w\| \leq 1$, we have $|z(x) - h(x)|^2 = O\left(\frac{1}{t} \ln \frac{t}{\delta}\right)$ by Lemma 13.

What is left is to check the mini-batch sizes; see the assumptions in Lemma 12 and Lemma 15. We need $\lambda_k(\mathbb{E}_i \left[h_i(x_i)^\top h_i(x_i)\right]) = \lambda_k(\mathbb{E}_x \left[h_i(x)^\top h_i(x)\right]) \pm O(1)$, so we only need to estimate $\mathbb{E}_x \left[h_i^j(x)^\top h_i^\ell(x)\right]$ up to constant accuracy for all $1 \leq j, \ell \leq k$, for which $B_{x,i} = O(\ln \frac{t}{\delta})$ suffices. We also need $\Delta_\omega = O(\lambda_k - \lambda_{k+1}) = O(1)$, so we only need $\Delta_\omega = O(1)$. This is a bound for $(tB_{x,i})^2$ pairs of points, for which $B_{\omega,i} = O(\ln \frac{t}{\delta})$ suffices. $\square$

Similar bounds hold for other random features, where the batch sizes will depend on the concentration bound of the random features used.

The rest of this section is the proof of the theorem. For simplicity, $\|\cdot\|_{\mathcal{F}}$ is shorten as $\|\cdot\|$.

First, we bound the difference between $H_t$ and $\tilde{G}_t$.

**Lemma 12.** *Suppose $|k(x, x')| \leq \kappa, |\phi_\omega(x)| \leq \phi$. Suppose the mini-batch sizes are large enough so that $\left|k(x_i, x_j) - \sum_{s=1}^{B_{\omega,i}} \phi_{\omega_s}(x_i) \phi_{\omega_s}(x_j)/B_{\omega,i}\right| \leq \Delta_\omega$ for all sampled data points $x_i$ and $x_j$. For any $w$ and $x$, with probability $\geq 1 - \delta$ over $(\mathcal{D}^t, \omega^t)$,*

$$|\tilde{g}_{t+1}(x)w - h_{t+1}(x)w|^2 \leq B_{t+1}^2 := \frac{1}{2}\Delta_\omega^2 \ln\left(\frac{2}{\delta}\right) \sum_{i=1}^{t} |\mathbb{E}_i |h_i(x_i)| a_{t,i} w|^2$$

*where $a_{t,i} = \eta_i \prod_{j=i+1}^{t} \left(I - \eta_j \mathbb{E}_j \left[h_j(x_j)^\top h_j(x_j)\right]\right)$ for $1 \leq i \leq t$, and $|h_i(x_i)| := \left[\left|h_i^j(x_i)\right|\right]_{j=1}^{k}$.*

*Furthermore, for any x and w,*

$$\mathbb{E}_{\mathcal{D}^t,\omega^t}|\tilde{g}_{t+1}(x)w - h_{t+1}(x)w|^2 \leq B_{2,t+1}^2 := \Delta_\omega^2 \sum_{i=1}^t \left|\mathbb{E}_i\left|h_i(x_i)\right|a_{t,i}w\right|^2.$$

*Proof.* Note that

$$H_{t+1} = \sum_{i=1}^t \mathbb{E}_i\left[\phi_{\omega_i}(x_i)\phi_{\omega_i}(\cdot)h_i(x_i)\right]a_{t,i} + F_0 a_{t,0}, \tag{49}$$

$$\tilde{G}_{t+1} = \sum_{i=1}^t \mathbb{E}_i\left[k(x_i,\cdot)h_i(x_i)\right]a_{t,i} + F_0 a_{t,0}, \tag{50}$$

where $a_{t,0} = \prod_{j=1}^t \left(I - \eta_j\mathbb{E}_j\left[h_j(x_j)^\top h_j(x_j)\right]\right)$.

We have $\tilde{g}_{t+1}(x)w - h_{t+1}(x)w = \sum_{i=1}^t V_{t,i}(x)$ where

$$V_{t,i}(x) = \mathbb{E}_i\left[k(x_i,x)h_i(x_i) - \phi_{\omega_i}(x_i)\phi_{\omega_i}(x)h_i(x_i)\right]a_{t,i}w.$$

$V_{t,i}(x)$ is a function of $(\mathcal{D}^i,\omega^i)$ and

$$\mathbb{E}_{\mathcal{D}^i,\omega^i}\left[V_{t,i}(x)|\omega^{i-1}\right] = \mathbb{E}_{\mathcal{D}^i,\omega^{i-1}}\mathbb{E}_{\omega_i}\left[V_{t,i}(x)|\omega^{i-1}\right] = 0,$$

so $\{V_{t,i}(x)\}$ is a martingale difference sequence.

Since $|V_{t,i}(x)| < \Delta_\omega|\mathbb{E}_i\left|h_i(x_i)\right|a_{t,i}w|$, the lemma follows from Azuma's Inequality. □

So to bound $|\tilde{g}_t(x)w - h_t(x)w|$, we need to bound $|\mathbb{E}_i\left|h_i(x_i)\right|a_{t,i}w|$, which requires some additional assumptions.

**Lemma 13** (Complete version of Lemma 7). *Suppose the conditions in Lemma 12 are true. Further suppose for all $i < t$, $\eta_i = \theta/i$ where $\theta$ is sufficiently large so that $\theta \geq 1/\lambda_k(\mathbb{E}_i\left[h_i(x_i)^\top h_i(x_i)\right])$; also suppose $\lambda_1\left(\tilde{G}_i^\top \tilde{G}_i\right) = O(1)$.*

*(1) With probability $\geq 1 - \delta$ over $(\mathcal{D}^t,\omega^t)$, for all $1 \leq i \leq t$ and $\ell \in [k]$, we have*

$$|\tilde{g}_i^\ell(x_i) - h_i^\ell(x_i)|^2 = O\left(\frac{\Delta_\omega^2\theta^4}{t}\ln\left(\frac{t}{\delta}\right)\right).$$

*(2) For any $x$ and unit vector $w$, with probability $\geq 1 - \delta$ over $(\mathcal{D}^t,\omega^t)$,*

$$|\tilde{g}_t(x)w - h_t(x)w|^2 = O\left(\frac{\Delta_\omega^2\theta^4}{t}\ln\left(\frac{t}{\delta}\right)\right).$$

*(3) For any $x$ and unit vector $w$,*

$$\mathbb{E}_{\mathcal{D}^t,\omega^t}|\tilde{g}_t(x)w - h_t(x)w|^2 = O\left(\frac{\Delta_\omega^2\theta^4\ln t}{t}\right).$$

*Proof.* We first do induction on statement (1), which is true initially. Assume it is true for $t$, we prove it for $t+1$.

We have that for any unit vector $w$,

$$|\mathbb{E}_i\,|h_i(x_i)|\,a_{t,i}w| = \left|\eta_i \mathbb{E}_i\,|h_i(x_i)| \prod_{j=i+1}^{t} \left[I - \eta_j \mathbb{E}_j\left[h_j(x_j)^\top h_j(x_j)\right]\right] w\right|$$

$$\leq \eta_i\,\|\mathbb{E}_i\,|h_i(x_i)|\|\,\|w\| \prod_{j=i+1}^{t} \left\|I - \eta_j \mathbb{E}_j\left[h_j(x_j)^\top h_j(x_j)\right]\right\|$$

$$\leq O(1)\frac{\theta^2}{i} \prod_{j=i+1}^{t} \left(1 - \frac{1}{j}\right) = O\left(\frac{\theta^2}{t}\right).$$

We use in the second line

$$\|h_i(x_i)\| \leq O\left(\sqrt{\frac{\theta^2}{t}\ln\frac{t}{\delta}}\right) + \|\tilde{g}_i(x_i)\| \leq O\left(\sqrt{\frac{\theta^2}{t}\ln\frac{t}{\delta}}\right) + \sqrt{\left\|\tilde{G}_i^\top \tilde{G}_i\right\|}\,\|\phi(x_i)\| = O(\theta)$$

that holds with probability $1 - t\delta/(t+1)$ by induction, and we use in the last line $\theta\lambda_k(\mathbb{E}_i\left[h_i(x_i)^\top h_i(x_i)\right]) \geq 1$.
Then by Lemma 12, with probability $\geq 1 - \delta/(k(t+1))$,

$$|\tilde{g}_{t+1}(x_{t+1})w - h_{t+1}(x_{t+1})w|^2 \leq \frac{1}{2}\Delta_\omega^2 \ln\left(\frac{2(t+1)}{\delta}\right) \sum_{i=1}^{t} |\mathbb{E}_i\,|h_i(x_i)|\,a_{t,i}w|^2$$

$$\leq O(\Delta_\omega^2)\ln\left(\frac{t+1}{\delta}\right) \sum_{i=1}^{t} \frac{\theta^4}{t^2} = O\left(\frac{\Delta_\omega^2\theta^4}{t+1}\ln\left(\frac{t+1}{\delta}\right)\right).$$

Repeating the argument for $k$ basis vectors $w = e_i(1 \leq i \leq k)$ completes the proof.
The other statements follow from similar arguments. $\qquad\square$

Next, we bound the difference between $\tilde{G}_t$ and $V$.

**Lemma 14.** *Suppose the conditions in Lemma 13 are true and furthermore, $\lambda_k(\tilde{G}_i^\top \tilde{G}_i) = \Omega(1)$ for all $i \in [t]$. Let $c_t^2$ denote $\cos^2\theta(\tilde{G}_t, V)$. Then with probability $\geq 1 - \delta$,*

$$c_{t+1}^2 \geq c_t^2\left\{1 + 2\eta_t\left[\lambda_k - \lambda_{k+1} - 2\|A_t - A\| - O\left(\Delta_\omega\theta^2\sqrt{\frac{1}{t}\ln\frac{t}{\delta}}\right)\right](1 - c_t^2) - O\left(\eta_t\Delta_\omega\theta^2\sqrt{\frac{1-c_t^2}{t}\ln\frac{t}{\delta}}\right)\right\} - \beta_t$$

*where $\beta_t$ is as defined in Lemma 8.*

*Proof.* The potential of $\tilde{G}_t$ can be computed by a similar argument as in the previous section; the only difference is replacing $A_t u$ with $k(x_t,\cdot)h_t(x_t)\hat{w}$. This leads to

$$\cos^2\theta(\tilde{G}_{t+1}, V) \geq c^2 + 2\eta_t u^\top\left(s^2 VV^\top - c^2 V_\perp V_\perp^\top\right) k(x_t,\cdot)h_t(x_t)\hat{w} - \beta_t$$

$$= c^2 + 2\eta_t u^\top\left(s^2 VV^\top - c^2 V_\perp V_\perp^\top\right)\left[(k(x_t,\cdot)h_t(x_t)\hat{w} - A_t u) + (A_t u - Au) + Au\right] - \beta_t \quad (51)$$

where $u = \tilde{G}_t\hat{w}$ with unit norm $\|u\| = 1$.
The terms involving $(A_t u - Au)$ and $Au$ can be dealt with as before, so we only need to bound the extra term

$$u^\top\left(s^2 VV^\top - c^2 V_\perp V_\perp^\top\right)\left[k(x_t,\cdot)h_t(x_t)\hat{w} - A_t u\right]$$

$$= u^\top\left(s^2 VV^\top - c^2 V_\perp V_\perp^\top\right)\left[k(x_t,\cdot)h_t(x_t)\hat{w} - k(x_t,\cdot)\tilde{g}_t(x_t)\hat{w}\right]$$

$$= u^\top\left(s^2 VV^\top - c^2 V_\perp V_\perp^\top\right) k(x_t,\cdot)[h_t(x_t) - \tilde{g}_t(x_t)]\hat{w}.$$

So we need to bound $[h_t(x_t) - \tilde{g}_t(x_t)]\widehat{w}$, which in turn relies on Lemma 13(1). More precisely, we have $\|h_t(x_t) - \tilde{g}_t(x_t)\|_\infty \leq \tilde{O}\left(\Delta_\omega \theta^2 \sqrt{1/t}\right)$ with probability $\geq 1 - \delta$. Also, we have $u = \tilde{G}_t \widehat{w}$ has unit norm, so $\|\widehat{w}\| = O(1)$ when $\lambda_k(\tilde{G}_i^\top \tilde{G}_i) = \Omega(1)$. Then

$$\left|u^\top VV^\top k(x_t, \cdot)[h_t(x_t) - \tilde{g}_t(x_t)]\widehat{w}\right| \leq \left\|u^\top V\right\| \left\|k(x_t, \cdot)\right\| \tilde{O}\left(\Delta_\omega \theta^2 \sqrt{1/t}\right) \leq c^2 \tilde{O}\left(\Delta_\omega \theta^2 \sqrt{1/t}\right)$$

where the last step follows from $c \geq 1/2$ by assumption. Similarly,

$$\left|u^\top V_\perp V_\perp^\top k(x_t, \cdot)[h_t(x_t) - \tilde{g}_t(x_t)]\widehat{w}\right| \leq \left\|u^\top V_\perp\right\| \left\|k(x_t, \cdot)\right\| \tilde{O}\left(\Delta_\omega \theta^2 \sqrt{1/t}\right) \leq s\tilde{O}\left(\Delta_\omega \theta^2 \sqrt{1/t}\right) = \tilde{O}\left(\Delta_\omega \theta^2 \sqrt{\frac{1 - c^2}{t}}\right).$$

Plugging into (51) and apply a similar argument as in Lemma 8 and 9 we have the lemma. $\square$

**Lemma 15** (Complete version of Lemma 6)**.** *If the mini-batch sizes are large enough so that $\|A - A_i\| < (\lambda_k - \lambda_{k+1})/8$, $\lambda_k(\mathbb{E}_i\left[h_i(x_i)^\top h_i(x_i)\right]) = \lambda_k(\mathbb{E}_x\left[h_i(x)^\top h_i(x)\right]) \pm O(1)$, and $\Delta_\omega = O(\lambda_k - \lambda_{k+1})$, then*

(1) $\theta = O(1)$;

(2) $1 - c_t^2 = O\left(\frac{1}{t} \ln \frac{t}{\delta}\right)$.

*Proof.* If the mini-batch size is large enough so that $\lambda_k(\mathbb{E}_i\left[h_i(x_i)^\top h_i(x_i)\right]) = \lambda_k(\mathbb{E}_x\left[h_i(x)^\top h_i(x)\right]) \pm O(1)$, we only need to show $\lambda_k(\mathbb{E}_x\left[h_i(x)^\top h_i(x)\right]) = \Omega(1)$, which will lead to $\theta = O(1)$ and then solving the recurrence in Lemma 14 leads to $1 - \cos^2\theta(\tilde{G}_{t+1}, V) = \tilde{O}(1/t)$.

Formally, we prove our statements (1)(2) by induction. They are true initially. Suppose they are true for $t - 1$, we prove them for $t$.

First, by solving the recurrence for $c_t$, we have that statement (2) is true up to step $t$.

Next, since $\mathbb{E}_x\left[\tilde{g}_t(x)^\top \tilde{g}_t(x)\right] = \tilde{G}_t^\top A\tilde{G}_t$, we have

$$\begin{aligned}
w^\top \mathbb{E}_x\left[\tilde{g}_t(x)^\top \tilde{g}_t(x)\right] w &= w^\top \tilde{G}_t^\top A\tilde{G}_t w \\
&= w^\top \tilde{G}_t^\top (V\Lambda_k V^\top + V_\perp \Lambda_\perp V_\perp^\top)\tilde{G}_t w \\
&\geq w^\top \tilde{G}_t^\top V\Lambda_k V^\top \tilde{G}_t w \\
&\geq \lambda_k c_t^2 \|w\|^2
\end{aligned}$$

which means $\lambda_k(\mathbb{E}_x\left[\tilde{g}_t(x)^\top \tilde{g}_t(x)\right]) = \Omega(1)$ by induction on $c_t$. This then leads to $\lambda_k(\mathbb{E}_i\left[h_i(x_i)^\top h_i(x_i)\right]) = \Omega(1)$, which means $\theta = O(1)$ up to step $t$. To see this, let $e_i(x) = h_i(x) - \tilde{g}_i(x)$. Then

$$\mathbb{E}_x\left[h_i(x)^\top h_i(x)\right] = \mathbb{E}_x\left[\tilde{g}_i(x)^\top \tilde{g}_i(x)\right] + 2\mathbb{E}_x\left[e_i(x)^\top h_i(x)\right] - \mathbb{E}_x\left[e_i(x)^\top e_i(x)\right].$$

By Lemma 13(3), $\mathbb{E}_x\left|e_i^j(x)\right| = \tilde{O}(\theta^4/t)$, which is $o(1)$ if $\theta = O(1)$. Then the norm of $2\mathbb{E}_x\left[e_i(x)^\top h_i(x)\right] - \mathbb{E}_x\left[e_i(x)^\top e_i(x)\right]$ is $o(1)$, so $\lambda_k(\mathbb{E}_x\left[\tilde{g}_i(x)^\top \tilde{g}_i(x)\right]) = \Omega(1)$ means $\lambda_k(\mathbb{E}_i\left[h_i(x_i)^\top h_i(x_i)\right]) = \Omega(1)$. $\square$

# 5  Extensions

The proposed algorithm is a general technique for solving eigenvalue problems in the functional space. Numerous machine learning algorithms boil down to this fundamental operation. Therefore, our method can be easily extended to solve many related tasks, including latent variable estimation, kernel CCA, *etc.*.

**Algorithm 3:** $\{\alpha_i, \beta_i\}_1^t = \textbf{DSGD-KSVD}(\mathbb{P}(x), \mathbb{P}(y), k)$

---

**Require:** $\mathbb{P}(\omega)$, $\phi_\omega(x)$.

1: **for** $i = 1, \ldots, t$ **do**
2:     Sample $x_i \sim \mathbb{P}(x)$. Sample $y_i \sim \mathbb{P}(y)$.
3:     Sample $\omega_i \sim \mathbb{P}(\omega)$ with seed $i$.
4:     $u_i = \textbf{Evaluate}(x_i, \{\alpha_j\}_{j=1}^{i-1}) \in \mathbb{R}^k$.
5:     $v_i = \textbf{Evaluate}(y_i, \{\beta_j\}_{j=1}^{i-1}) \in \mathbb{R}^k$.
6:     $W = u_i v_i^\top + v_i u_i^\top$
7:     $\alpha_i = \eta_i \phi_{\omega_i}(x_i) v_i$.
8:     $\beta_i = \eta_i \phi_{\omega_i}(y_i) u_i$.
9:     $\alpha_j = \alpha_j - \eta_i W \alpha_j$, for $j = 1, \ldots, i-1$.
10:    $\beta_j = \beta_j - \eta_i W \beta_j$, for $j = 1, \ldots, i-1$.
11: **end for**

## 5.1 Locating individual eigenfunctions

The proposed algorithm finds the subspace spanned by the top $k$ eigenfunctions, but it does not isolate the individual eigenfunctions. When we need to locate these individual eigenfunctions, we can use a modified version. Its update rule is

$$G_{t+1} = G_t + \eta_t A_t G_t - \eta_t G_t \, \text{UT}\left[G_t^\top A_t G_t\right], \tag{52}$$

where $\text{UT}[\cdot]$ is an operator that sets the lower triangular parts to zero.

To understand the effect of the upper triangular operator, we can see that $\text{UT}[\cdot]$ forces the update rule for the first function of $G_t$ to be exactly the same as that of one-dimensional subspace; all the contributions from the other functions are zeroed out.

$$g_{t+1}^1 = g_t^1 + \eta_t A_t g_t^1 - \eta_t g_t^1 {g_t^1}^\top A_t g_t^1, \tag{53}$$

Therefore, the first function will converge to the eigenfunction corresponding to the top eigenvalue.

For all the other functions, $\text{UT}[\cdot]$ implements a Gram-Schmidt-like orthogonalization that subtracts the contributions from other eigenfunctions.

## 5.2 Latent variable models and kernel SVD

Our algorithm can be straightforwardly extended to solve kernel SVD. The extension hinges on the following relation

$$\begin{bmatrix} 0 & A^\top \\ A & 0 \end{bmatrix} \begin{bmatrix} V \\ U \end{bmatrix} = \begin{bmatrix} A^\top U \\ AV \end{bmatrix} = \begin{bmatrix} V \\ U \end{bmatrix} \Sigma,$$

where $U\Sigma V^\top$ is the SVD of $A$.

It is therefore reduced to the eigenvalue problem. Plugging it into the update rule and treating the two blocks separately, we thus get two simultaneous update rules

$$W_t = U_t^\top A V_t + V_t^\top A^\top U_t \tag{54}$$

$$U_{t+1} = U_t + \eta_t \left(AV_t - U_t W_t\right), \tag{55}$$

$$V_{t+1} = V_t + \eta_t \left(A^\top U_t - V_t W_t\right). \tag{56}$$

The algorithm for updating the coefficients is summarized in Algorithm 3.

**Algorithm 4:** $\{\alpha_i, \beta_i\}_1^t = \textbf{DSGD-KCCA}(\mathbb{P}(x), \mathbb{P}(y), k)$

---

**Require:** $\mathbb{P}(\omega)$, $\phi_\omega(x)$.

1: **for** $i = 1, \ldots, t$ **do**
2:    Sample $x_i \sim \mathbb{P}(x)$. Sample $y_i \sim \mathbb{P}(y)$.
3:    Sample $\omega_i \sim \mathbb{P}(\omega)$ with seed $i$.
4:    $u_i = \textbf{Evaluate}(x_i, \{\alpha_j\}_{j=1}^{i-1}) \in \mathbb{R}^k$.
5:    $v_i = \textbf{Evaluate}(y_i, \{\beta_j\}_{j=1}^{i-1}) \in \mathbb{R}^k$.
6:    $W = u_i v_i^\top + v_i u_i^\top$
7:    $\alpha_i = \eta_i \phi_{\omega_i}(x_i) [v_i - W u_i]$.
8:    $\beta_i = \eta_i \phi_{\omega_i}(y_i) [u_i - W v_i]$.
9: **end for**

---

## 5.3 Kernel CCA and generalized eigenvalue problem

Kernel CCA and ICA can also be solved under the proposed framework because they can be viewed as generalized eigenvalue problem. Given two variables $X$ and $Y$, CCA finds two projections such that the correlations between the two projected variables are maximized. Given the covariance matrices $C_{XX}$, $C_{YY}$, and $C_{XY}$, CCA is equivalent to the following problem

$$\begin{bmatrix} C_{XX} & C_{XY} \\ C_{YX} & C_{YY} \end{bmatrix} \begin{bmatrix} g_X \\ g_Y \end{bmatrix} = (1 + \sigma^2) \begin{bmatrix} C_{XX} & \\ & C_{YY} \end{bmatrix} \begin{bmatrix} g_X \\ g_Y \end{bmatrix},$$

where $g_X$ and $g_Y$ are the top canonical correlation functions for variables $X$ and $Y$, respectively, and $\sigma$ is the corresponding canonical correlation.

This is a generalized eigenvalue problem. It can reformulated as the following non-convex optimization problem

$$\max_G \operatorname{tr}\left(G^\top A G\right), \tag{57}$$

$$\text{s.t. } G^\top B G = I. \tag{58}$$

Following the derivation for the standard eigenvalue problem, we get the foliowing update rules

$$G_{t+1} = G_t + \eta_t \left(I - BG_t G_t^\top\right) A G_t. \tag{59}$$

Denote $G_t^X$ and $G_t^Y$ the canonical correlation functions for $X$ and $Y$, respectively. We can rewrite the above update rule as two simultaneous rules

$$W_t = G_t^{Y\top} C_{YX} G_t^X + G_t^{X\top} C_{XY} G_t^Y \tag{60}$$

$$G_{t+1}^X = G_t^X + \eta_t \left[C_{XY} G_t^Y - C_{XX} G_t^X W\right] \tag{61}$$

$$G_{t+1}^Y = G_t^Y + \eta_t \left[C_{YX} G_t^X - C_{YY} G_t^Y W\right]. \tag{62}$$

We present the detailed updates for coefficients in Algorithm 4.