[Reviews · NeurIPS 2015]

Submitted by Assigned_Reviewer_1

This papers deals with kernel component analysis for massive data sets. More specifically, the authors propose the use of both random features of kernel functions and stochastic gradient descent to speed up nonlinear components analysis methods such as kernel PCA and kernel CCA. The authors provide a convergence analysis of the method and perform experiments on synthetic and real data sets showing that their method is week suited for large scale data.

The paper is well written. It builds upon recent works on scaling up kernel methods to large scale data sets. The main idea of the paper is not novel. It combines random features for component analysis ([12] ICML'14) with doubly stochastic gradients ([6] NIPS'14). The main contribution is the extension of the convergence results in [6] to the non-convex case.
Summary: A well-written paper which shows how to apply the doubly stochastic gradient method to kernel component analysis in order to make them more practical with large data sets. The novelty is somewhat limited.

Submitted by Assigned_Reviewer_2

This paper proposes a computationally efficient way of scaling kernel non-linear component

analysis. Previous attempts of scaling kernel PCA-like eigenproblems in machine learning

resorted to Nystrom-variant subsampling or random feature linearization of the problem. Each of

these methods works effectively in certain settings but the number of samples/features may be

prohibitively large in some settings.

This paper proposes a doubley-stochastic gradient algorithm for solving kernel eigenvalue problems encompassing kernel PCA, CCA, and SVD. It is called doubly-stochastic because sampling is performed in the data samples and the random features (for this paper, limited to

stationary kernels whose Fourier transform is well-defined). The paper claims convergence rate of $\tilde{O}(1/t)$ to the global optimum for the recovered eigensubspace (in terms of principal angle), but with a caveat being that the step sizes are chosen properly and the mini-batch size is sufficiently large.

Quality: The paper is reasonably well-written with no glaring grammar errors/typos.

Clarity: Clear in most parts, some suggestions for improvement: - In Line 300: "which outside the RKHS" -> "which is outside the RKHS" - Please provide the citation for the "median trick" - just to make the paper self-contained. - Please provide the dimensionality of the datasets used in the experiment. For example, the synthetic dataset is mentioned without such information. - Figure 1 and Figure 2 and Figure 3 are hard to look at, when printed on the paper. - Equation (6), Line 305: the Part II should read $ 2 | \tilde{g}_t(x) - h_t(x) |^2 $

Originality: The idea of doubly-stochastic gradient descent has appeared in the convex setting where the objective function can be expressed as a convex function (kernel SVM, etc.). This paper attempts to prove the global convergence rate for the principal angle bound on the recovered kernel eigensubspace.

Significance: Kernel eigenvalue problems are notoriously hard to scale and achieving high accuracy is challenging. Addressing this problem is a worthy effort.
Summary: An interesting paper addressing the scalability issue for kernel eigenvalue problems arising in machine learning methods. An overall good paper with some theoretical results.

Submitted by Assigned_Reviewer_3

Strong contribution. Highly useful. Seems easy to implement. Non-trivial convergence analysis. Large-scale experiments.

However, I have some concerns that the experiments might not be completely fair:

- In the experiments, the authors systematically use more features for their method than for the Random Fourier and the Nystroem methods

- In Figure 4, why not show the results of RF features at 10240 features? and the results of DSGD for 512 to 4096 features?

Minor remark: in the related work section, the authors could briefly mention the Nystroem-based approaches.
Summary: Strong, highly useful contribution. Some concerns regarding the fairness of the experiments.

Submitted by Assigned_Reviewer_4

Scaling up unsupervised learning algorithms like kernel PCA, kernel SVD is very important problem, regarding the large amount of applications they could be used for.

Inspired by the previous works on online algorithm for PCA, they propose an algorithm for kernel PCA using doubly stochastic gradients with randomness from both batched stochastic gradient and random Fourier features. The doubly stochastic gradients has been introduced for support vector machine previously, but here the setting is slightly different because of the orthogonality constraint in component analysis. Overall, I think its a nice connection between the online algorithm for PCA and doubly stochastic gradient descent.

The analysis tries to characterize the convergence in terms of the principal angle between the sub-spaces. The assumption on a good initialization seems very strong to me. I would like to see more discussion about this condition: for example, given a very high dimensional space like RKHS, how likely this assumption will be satisfied for a simple initialization algorithm, such as randomly draw $k$ points to form a subspace. Another issue I have with the analysis is that the required batch size seems to increase as $t$ increases, even though it is only increasing logarithmically. This fact seems to downgrade the applicability of the algorithm in practice?

Regardless of the minor issues in the analysis, the paper contains comprehensive and interesting experiments to demonstrate the effectiveness of the algorithm.
Summary: This paper tries to scale up the kernel-based nonlinear component analysis, using the doubly stochastic gradients. The algorithm is very close to the doubly stochastic gradients for support vector machine, while the underlying analysis seems different.

Author Feedback
Author rebuttal: We would like to thank the reviewers and area chairs for the time and efforts in the review process.

Reviewer 1:

Thank you for the positive comments. We will fix the typos and fill in more details in the revision.

Reviewer 2:

We used more random features for the proposed algorithm because it is the only algorithm that can scale up to tens of thousands or even more random features. Vanilla batch random features and Nystrom methods are limited to using smaller number of random features (<10K) due to the quadratic memory requirement. Indeed, one of the advantages of our algorithm is the ability to use many more random features without explosive memory issues.

We will add more description of Nystrom-based approaches in the revision.

Reviewer 3:

As for initialization, a simple approach works like this: take a small set of m data points, and a small set of m random features corresponding to the kernel, and run batch PCA in the random feature space. The size m can be thousands such that batch PCA can be carried out in most modern desktop computers. After that one runs the doubly stochastic gradient to refine this solution with more data points and random features. This initialization gets better as we increase m, and it is similar to the random nonlinear component method in [1].

[1] D. Lopez-Paz, S. Sra, A. Smola, Z. Ghahramani, and B. Scholkopf. Randomized nonlinear component analysis. ICML'14.

The dimension in kernel PCA should be interpreted differently from that in the finite dimension PCA. It depends on the property of the kernel and its interaction with the input distribution. This is similar to the Rademachar complexity for RKHS function which does not explicitly depend on the dimension of the input data. More specifically, the difficulty level in the kernel case is reflected in the spectrum decay of the covariance operator. By taking into account the spectrum like in [2], one can obtain a more refined analysis of our algorithm and possibly even better convergence rate.

[2] F. Bach. On the Equivalence between Quadrature Rules and Random Features. Technical report, HAL-01118276, 2015.

As for the batch size growing with log(t), it comes from a union bound over the probability that the mini-batch update fails to be estimated up to constant error for all t iterations. It could be an artifact of the analysis. In practice, we observe that it has little negative effect by using a fixed batch size. In the experiments, we often take batch sizes to be as large as hundreds or thousands. It is likely in these cases, the conditions needed in the theory are already satisfied.

Review 4, 5, 6:

First, we would like to clarify the difference between the current paper and the convex version [6]. Although the algorithmic procedures appear similar, the analysis for the convex case do not easily translate to the eigenvalue problems. More specifically, the nonlinear component analysis involves additional challenges:

I. We prove the convergence rate for a non-convex problem, which requires making use of the special structures of the eigenvalue problem. Such structure is very different from the convex case.

II. Orthogonality constraints for the components are challenging to enforce in the large scale kernel case, as a result Gram-Schmidt orthogonalization is not applicable here. Instead, we use a Lagrangian multiplier approach, for which we show that it gradually enforces the constraints.

III. We analyze the convergence of principle angles between subspaces which are novel and not present in the convex case.

IV. We have identified the relation between the batch size, the eigen-gap, and iteration number, which are novel and not present in the convex case.

As for regenerating the random features, one only needs to remember the seed for the random number generator.